# Analyzing the Training Dynamics of Image Restoration Transformers: A Revisit to Layer Normalization

**MinKyu Lee,  Sangeek Hyun,  Woojin Jun,  Hyunjun Kim,  Jiwoo Chung,  Jae-Pil Heo***

Sungkyunkwan University

{2minkyulee, hse1032, junwoojinjin, arithu3, jiwoo.jg, jaepilheo}@gmail.com

https://github.com/2minkyulee/i-LN

## Abstract

This work analyzes the training dynamics of Image Restoration (IR) Transformers and uncovers a critical yet overlooked issue: conventional LayerNorm (LN) drives feature magnitudes to diverge to a *million scale* and collapses channel-wise entropy. We analyze this in the perspective of networks attempting to bypass LN's constraints that conflict with IR tasks. Accordingly, we address two misalignments between LN and IR: 1) *per-token normalization* disrupts spatial correlations, and 2) *input-independent scaling* discards input-specific statistics. To address this, we propose Image Restoration Transformer Tailored Layer Normalization ($i$-LN), a simple drop-in replacement that normalizes features holistically and adaptively rescales them per input. We provide theoretical insights and empirical evidence that this simple design effectively leads to both improved training dynamics and thereby improved performance, validated by extensive experiments.

## 1 Introduction

Image restoration (IR) aims to reconstruct high-quality images from degraded inputs. With the success of Vision Transformers (Dosovitskiy et al., 2020), Transformer-based architectures coupled with LayerNorm (LN) have been actively adopted for IR tasks and are now a common standard (Liang et al., 2021; Chen et al., 2023a; Hsu et al., 2024). However, despite recent architectural advances, the underlying training dynamics of IR Transformers remain underexplored.

This inspires us to take a closer look at their internal behavior, leading us to uncover a critical yet overlooked phenomenon: feature magnitudes diverge dramatically, reaching scales up to a *million*, while channel-wise feature entropy drops sharply (Fig.1). Interestingly, this phenomenon aligns with previous studies (Karras et al., 2020; Wang et al., 2022a), which similarly observed visual artifacts and abnormal features when coupled with specific normalization layers. However, discussions specific to the unique requirements of IR tasks and IR Transformers were not made.

Building on these insights, we hypothesize that the observed feature divergence in IR Transformers arises from networks attempting to circumvent LN, due to constraints of LN that do not align with the unique requirements of IR tasks. Accordingly, we identify two key mismatches between LayerNorm and IR tasks; supported by both theoretical insights and extensive empirical analysis.

First, LayerNorm operates in a per-token manner, without considering inter-pixel (token) relationships. This disrupts the spatial correlations in input features, an aspect crucial for high-fidelity image restoration. Second, it maps intermediate features into a unified normalized space, limiting the range flexibility of internal representations. This thereby disregards the input-dependent statistical variability (Lim et al., 2017b) that is inherent in IR tasks. Together, these mismatches significantly hinder IR Transformer's ability to accurately preserve low-level features throughout the network, which is necessary for faithful image restoration.

While one intuitive solution could be the complete removal of normalization layers as prior works have done (Lim et al., 2017b; Wang et al., 2018; Karras et al., 2020; 2024), our experimental observations

---

*Corresponding Author.

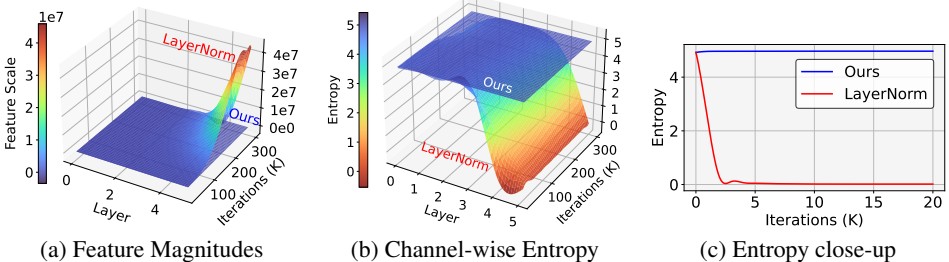

(a) Feature Magnitudes    (b) Channel-wise Entropy    (c) Entropy close-up

Figure 1: **Visualization of feature magnitudes and channel-wise entropy during training of an Image Restoration (IR) Transformer using conventional LayerNorm (LN) and $i$-LN (Ours). (a)** Evolution of feature magnitudes across layers and training iterations, highlighting the dramatic divergence (to million-scale) under conventional LN. **(b-c)** Channel-wise entropy with LN drops sharply at the very early stage of training, indicating the emergence of acute peaks hidden in specific channels. Ours $i$-LN exhibits well-distributed activation across channels and significant stability.

highlight significant training instability when normalization is entirely omitted from IR Transformers (Tab.1); the network fails to converge.

In this work, we show that these issues can be addressed in a surprisingly simple manner; leading to significant stability and substantial performance gain. We propose the *Image Restoration Transformer Tailored Layer Normalization* ($i$-LN), which acts as a drop-in replacement to conventional (vanilla) LayerNorm by better aligning with the unique requirements of IR tasks.

Instead of normalizing each token independently, we propose to apply normalization across the entire spatio-channel dimension within IR Transformers (Fig.3), effectively preserving spatial correlationships among tokens, contrary to vanilla per-token LayerNorm. Furthermore, We rescale features with the normalization parameters after each attention and feed-forward layer, explicitly enabling range flexibility and accounting for input-dependent variations in internal feature statistics. Together, these modifications effectively preserve low-level feature statistics throughout the network, better aligning with the requirements of IR tasks.

Extensive experiments show that $i$-LN leads to both stable training dynamics with improved performance across various IR benchmarks. Additionally, we observe cues suggesting robustness under reduced-precision configurations, RealWorld IR tasks, and improved spatial correlation modeling.

## 2 REVISITING LAYER NORMALIZATION IN IR TRANSFORMERS

**Observation (Abnormal Feature Statistics).** Our initial analysis focuses on tracking the trajectory of internal features during the training of IR Transformers. We visualize the squared mean of intermediate features at each basic building block of the network, following (Karras et al., 2024). We select the x4 SR task using the HAT (Chen et al., 2023a) model as the representative IR task (Fig.1).

The analysis reveals that feature statistics diverge dramatically, reaching values up to a *million* scale. To pinpoint the origins of this feature divergence, we analyze the feature entropy across the channel-axis. Analysis demonstrates a sharp decrease in feature entropy, which indicates the presence of channels with extreme values that dominate the statistics. Since these extreme values are unusual, this motivates us to further investigate. Accordingly, we analyze the training dynamics across configurations by varying the network scale (Fig.2a-2b), varying the IR tasks (Fig.2c), and varying the normalization scheme (Fig.4); and observe that this phenomenon occurs across all configurations utilizing standard IR Transformers. While this type of hidden abnormal behavior aligns with the observations in prior studies (Karras et al., 2020; Wang et al., 2018; 2022a), further discussion did not gain much attention, especially regarding the unique properties and requirements of IR Transformers.

In the following, we provide further insights into this phenomenon by examining the characteristics of LayerNorm (LN), the de facto normalization in IR Transformers. We start by defining the spatial relationship between pixels (i.e., inter-pixel structure), and further show that conventional LayerNorm cannot preserve this. For simplicity, we neglect the affine parameters for theoretical analysis.

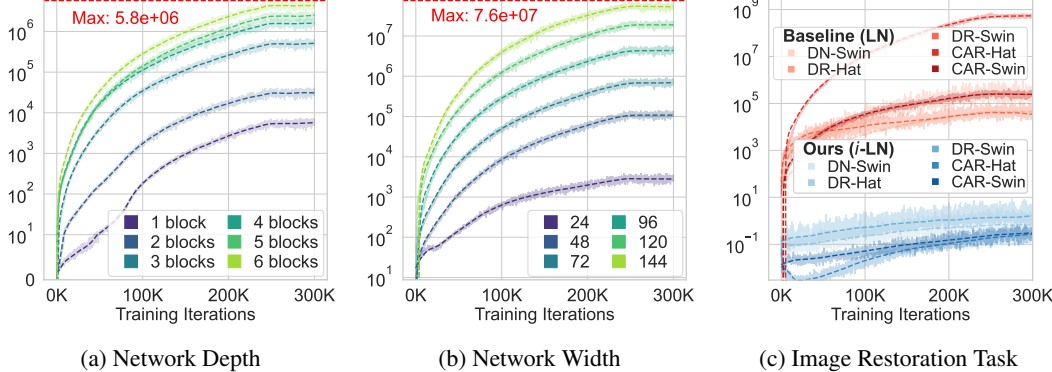

(a) Network Depth  (b) Network Width  (c) Image Restoration Task

Figure 2: **Feature magnitude evolution in IR Transformers across different settings. (a-b)** Feature divergence signifies as the network scales. **(c)** Feature divergence appears across various Transformer backbones and IR tasks: super-resolution (SR), denoising (DN), deraining (DR), JPEG compression artifact removal (CAR), demonstrating that this phenomenon is widespread. It can be effectively mitigated by simply replacing conventional LayerNorm with the proposed $i$-LN.

**Definition 1** (Inter-pixel Structure and Preservation). *Let $x \in \mathbb{R}^{L \times C}$ be a feature map with $L$ tokens. We write the $\ell$-th token as $x_\ell \in \mathbb{R}^C$ and the $c$-th element of it as $x_{\ell,c} \in \mathbb{R}$. The* inter-pixel structure *of a feature map is given by the set of relative differences $\Delta x := \{ x_\ell - x_k : 1 \le \ell, k \le L \}$.*

**Definition 2** (Structure Preserving Transformation). *A transformation $T$ is said to* preserve inter-pixel structure up to scale *if there exists a homothety $H(x) = ax + b$, with $a > 0$ and $b \in \mathbb{R}^C$, such that*

$$T(x_\ell) - T(x_k) = H(x_\ell - x_k) = a(x_\ell - x_k) \quad \text{for all } \ell, k.$$

*Such maps preserve all angles and pairwise distance ratios, and correspond to a single global shift and uniform scaling across all tokens. For $a = 1$, $T$ is said to preserve structure absolutely.*

Intuitively, consider $x$ as a point cloud in $\mathbb{R}^C$, where each point represents a token. A structure-preserving transformation may only uniformly scale and shift the entire cloud. That is, the overall shape of the point cloud should be preserved up to a single global scaling factor and translation.

**Vanilla Per-token LayerNorm (Baseline).** Conventional Transformer architectures utilize the per-token LayerNorm (LN) as the de facto normalization scheme which operates as follows:

$$\text{LN}(x_\ell) = \gamma \frac{1}{\sqrt{\sigma_\ell^2 + \epsilon}}(x_\ell - \mu_\ell) + \beta, \qquad \mu_\ell = \mathbb{E}_c[x_{\ell,c}], \qquad \sigma_\ell^2 = \mathbb{E}_c[(x_{\ell,c} - \mu_\ell)^2], \qquad (1)$$

where $\mathbb{E}_c[\cdot]$ is taken over the channel dimension $c$, and $\gamma, \beta \in \mathbb{R}^c$ are each affine parameters applied after the normalization step, and LN operates for each token $x_\ell$ given the entire input feature $x$.

**Proposition 1.** *(Vanilla LayerNorm fails to preserve structure). Let $T_{\text{LN}}$ be the normalization in vanilla per-token LN. Then, in general, there do not exist $a > 0$ and an orthogonal $Q$ such that*

$$T_{\text{LN}}(x_\ell) - T_{\text{LN}}(x_k) = a Q (x_\ell - x_k) \qquad \text{for all } x_\ell, x_k,$$

*Thus $T_{\text{LN}}$ is not even conformal on the token set. Since homotheties are strict subclasses of conformal maps, $T_{\text{LN}}$ is not a homothety and therefore it does not preserve inter-pixel structure in general.*

**Remark.** The exception arises in degenerate cases where all tokens share identical per-token mean and variance, in which case a similarity map can exist (i.e., inter-pixel structure is preserved). Such cases are extremely rare in practice. Our intuition is that since LN cannot naturally preserve inter-pixel structure, networks learn to generate large magnitude features regardless of the input, thereby, manipulate the overall feature statistic to behave similarly to this exceptional degenerate scenario.

Inspired by prior observations, we hypothesize that feature divergence arises from a fundamental mismatch between the requirements of IR tasks and the constraints imposed by LayerNorm, leading us to propose a tailored normalization scheme that aligns with the unique requirements of IR tasks.

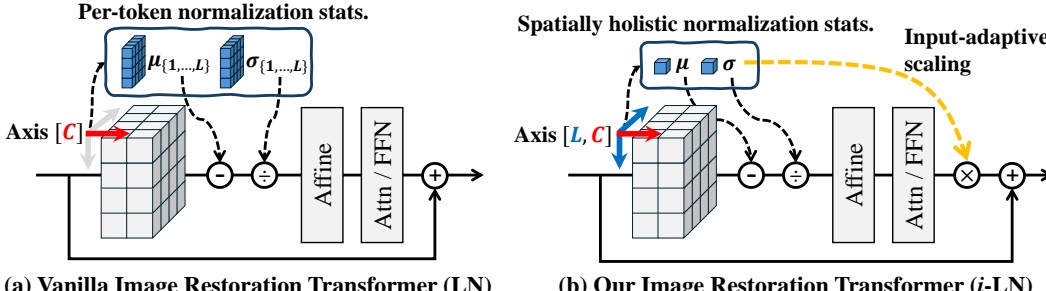

Figure 3: Comparison between IR Transformer blocks using conventional per-token LayerNorm (LN) and our proposed $i$-**LN**. Contrary to conventional LN, which normalizes each token independently, our $i$-LN applies holistic normalization across the entire spatio-channel dimension, preserving essential spatial correlations between tokens. Additionally, $i$-LN input-adaptively rescales features after the attention (Attn) and feedforward (FFN) layers, thereby better preserving input statistics and allowing feature range flexibility. These together enable IR Transformers to preserve low-level characteristics of input throughout the network, aligning with the unique requirements of IR.

## 3 METHOD: TAILORING LAYERNORM FOR IR TRANSFORMERS

**Spatial Holisticness in Normalization (LN*).** We propose a simple variant of LN that improves in preserving inter-token spatial relationships of input features, which we refer to as LN*. Instead of normalizing each token individually as LN, we derive normalization statistics from the entire spatio-channel dimension of the input feature as follows:

$$\text{LN}^*(x) = \gamma \frac{1}{\sqrt{\sigma^2 + \epsilon}}(x - \mu) + \beta, \qquad \mu = \mathbb{E}_{\ell,c}[x_{\ell,c}], \qquad \sigma^2 = \mathbb{E}_{\ell,c}[(x_{\ell,c} - \mu)^2], \qquad (2)$$

where the expectation $\mathbb{E}_{\ell,c}[\cdot]$ is taken over both spatial ($\ell$) and channel dimensions ($c$). This straightforward modification effectively mitigates the issue raised by the per-token operation in vanilla per-token LayerNorm. While normalization methods in CNNs already inherently work in a spatially holistic manner, the implications of such holisticness in normalization and the corresponding spatial structure corruption without it have received little attention, particularly in the context of IR Transformers. With this point, the following section aims to provide further intuition and establish connections between holisticness and spatial structure (i.e., inter-pixel structure) preservation.

**Proposition 2.** *(LN\* preserves structure). Let $T_{LN^*}$ be the normalization defined by LN\*, with global mean $\mu$ and std. $\sigma > 0$ computed over all tokens and channels. Then for any two tokens $x_\ell, x_k$,*

$$T_{LN^*}(x_\ell) - T_{LN^*}(x_k) = (1/\sigma)(x_\ell - x_k).$$

*Thus, $T_{LN^*}$ is a homothety, and accordingly, preserves spatial structure up to a global scale.*

**Remark.** In short, LN* is structure-preserving up to one missing scalar (i.e., the global scale). We handle this loss of information by explicitly reintroducing it later, as described below.

**Preserving Input Dependent Statistics.** We further tailor the normalization operator to better suit the requirements of IR tasks. Specifically, we address the issue of input-blind normalization. While IR tasks require the preservation of input-dependent feature statistics for faithful reconstruction, both conventional LayerNorm and even the holistic LN* overlooks this aspect by mapping features into a unified normalized space. Although normalization improves training stability, it also causes the model to lose critical input-dependent information (i.e., the missing global scale term of inter-pixel structure) by restricting the range flexibility of internal representations (Lim et al., 2017b).

Accordingly, we propose a simple input-adaptive rescaling strategy that effectively tackles this issue. We rescale the output of Attention and FFN by their standard deviation computed in the preceding normalization process as the yellow line in Fig.3b, which we refer to as $i$-LN. Accordingly, a typical Attention or FFN block $B$ could be further improved by coupling with $i$-LN as follows:

$$B(x; f, i\text{-LN}) = x + \sqrt{\sigma^2 + \epsilon} \cdot f(\text{LN}^*(x)), \qquad (3)$$

| Idx | Method | SH | Set14 | | BSD100 | | Urban100 | | Manga109 | |
|---|---|---|---|---|---|---|---|---|---|---|
| | | | PSNR | SSIM | PSNR | SSIM | PSNR | SSIM | PSNR | SSIM |
| 1 | LayerNorm | ✗ | 28.79 | .7876 | 27.68 | .7411 | 26.55 | .8015 | 31.01 | .9150 |
| 2 | LayerScale | ✗ | 28.89 | .7887 | 27.76 | .7426 | 26.75 | .8058 | 31.37 | .9178 |
| 3 | RMSNorm | ✗ | 28.88 | .7879 | 27.74 | .7417 | 26.67 | .8037 | 31.24 | .9165 |
| 4 | ReZero | ✓ | 28.81 | .7861 | 27.70 | .7406 | 26.41 | .7964 | 31.05 | .9147 |
| 5 | None | ✓ | - | - | - | - | - | - | - | - |
| 6 | InstanceNorm | ✓ | 28.98 | .7907 | 27.80 | .7445 | 27.02 | .8136 | 31.46 | .9199 |
| 7 | BatchNorm† | ✓ | 28.95 | .7901 | 27.80 | .7442 | 26.70 | .8123 | 31.39 | .9186 |
| 8 | $i$-LN (Ours) | ✓ | **29.01** | **.7915** | **27.84** | **.7456** | **27.17** | **.8167** | **31.82** | **.9228** |

Table 1: **Comparison between various normalization schemes**. † indicates that BatchNorm is evaluated in train-mode. SH indicates the spatial holisticness of the normalization scheme, including the setting without any normalization (None). Experiments are performed for $\times 4$ SR with $HAT_1$. The best result for each setting is highlighted in **bold**.

where $f$ is either the according Attention or FFN operation of block $B$. Overall, this reintroduces the original feature statistic lost due to normalization. This simple strategy enables IR Transformers to better preserve the per-instance statistics throughout the network and allows range flexibility to intermediate features. We later show that this leads to an order of magnitude more stable feature distribution (i.e., higher entropy) and overall improved IR performance.

**Remark.** This simple input-adaptive rescaling strategy explicitly reintroduces the missing global scaling term that LN* could not preserve (i.e., the restricted range flexibility issue).

## 4 EXPERIMENTS

**Training Settings.** Since recent works have discrepancies in their detailed training settings (Chen et al., 2024), we reimplement baseline methods and our method under identical settings for fair comparison. Networks for deraining (DR) were trained on Rain13K (Jiang et al., 2020), while DF2K (DIV2K (Agustsson & Timofte, 2017) + Flickr2K (Lim et al., 2017a)) was used for other tasks. Only basic augmentations (random flips, rotations, crops) were applied, without mixing augmentations, progressive patch sizing, or warm-start. SwinIR (Liang et al., 2021), HAT (Chen et al., 2023a), and DRCT (Hsu et al., 2024) were used as backbones. Configurations for each backbone model (e.g., model size, batch size, and patch size) are adjusted to meet computational constraints and are denoted by subscripts (e.g., $HAT_1$, $HAT_2$). The full specifications are provided in Appendix 5.

**Evaluation Settings.** Standard benchmarks are employed including: Set5 (Bevilacqua et al., 2012), Set14 (Zeyde et al., 2010), BSD100 (Martin et al., 2001), Urban100 (Huang et al., 2015), Manga109 (Matsui et al., 2017) for SR; CBSD68 (Martin et al., 2001), Kodak (Franzen, 1999), McMaster (Zhang et al., 2011), Urban100 for DN; LIVE1 (Sheikh, 2005), Classic5 (Foi et al., 2007), Urban100 (Huang et al., 2015) for CAR; Test100 (Zhang et al., 2019) and Rain100L (Yang et al., 2017) for DR. We crop Urban100 into non-overlapping 256×256 patches due to memory limits for CAR and DN. We report PSNR and SSIM indices. Experiments were performed on NVIDIA A6000s.

### 4.1 NORMALIZATION SCHEME VARIATION

We analyze the effects of various normalization techniques, including representative normalization schemes as vanilla Layer-Norm (LN) (Ba et al., 2016), per-token RMSNorm (RMS) (Zhang & Sennrich, 2019), InstanceNorm (IN) (Ulyanov et al., 2016), Batch-Norm (BN) (Ioffe & Szegedy, 2015), and our proposed $i$-LN. Considering previous studies where completely removing normalization from SR networks (Lim et al., 2017b; Wang et al., 2018) led to performance improvements, we additionally tested a similar configuration indicated as *None*, where normalizations are entirely removed.

Further, we investigate the empirical impacts of recent methods designed to stabilize Transformer training: ReZero (RZ) (Bachlechner et al., 2021) and LayerScale (LS) (Touvron et al., 2021). ReZero removes LayerNorm from Transformer blocks and multiplies a learn-

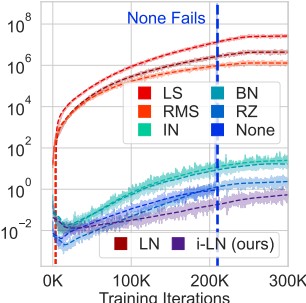

Figure 4: Feature divergence across various normalizations.

able zero-initialized scalar to the residual path. Similarly, LayerScale multiplies a near-zero-initialized learnable diagonal matrix to the residual path but reintroduces LayerNorm. Since both methods initially multiply a (near) zero-scale factor to the network output, we consider them as potential solutions

Table 2: Quantitative comparison between the conventional LayerNorm (LN) and our proposed $i$-LN across diverse IR tasks. The best result for each setting is highlighted in **bold**.

| Backbone | Scale | Set5 PSNR | Set5 SSIM | Set14 PSNR | Set14 SSIM | BSD100 PSNR | BSD100 SSIM | Urban100 PSNR | Urban100 SSIM | Manga109 PSNR | Manga109 SSIM |
|---|---|---|---|---|---|---|---|---|---|---|---|
| HAT$_1$ + LN | ×2 | 38.14 | .9610 | 33.78 | .9196 | 32.19 | .9000 | 32.16 | .9297 | 38.84 | .9778 |
| HAT$_1$ + $i$-LN | ×2 | **38.37** | **.9619** | **34.08** | **.9218** | **32.42** | **.9028** | **33.32** | **.9385** | **39.69** | **.9794** |
| DRCT$_1$ + LN | ×2 | 38.19 | .9613 | 33.28 | .9197 | 32.28 | .9010 | 32.60 | .9323 | 39.23 | .9785 |
| DRCT$_1$ + $i$-LN | ×2 | **38.23** | **.9614** | **33.86** | **.9206** | **32.31** | **.9014** | **32.79** | **.9344** | **39.40** | **.9788** |
| HAT$_1$ + LN | ×4 | 32.51 | .8992 | 28.79 | .7876 | 27.68 | .7411 | 26.55 | .8015 | 31.01 | .9150 |
| HAT$_1$ + $i$-LN | ×4 | **32.72** | **.9019** | **29.01** | **.7915** | **27.84** | **.7456** | **27.17** | **.8167** | **31.82** | **.9228** |
| DRCT$_1$ + LN | ×4 | 32.50 | .8989 | 28.85 | .7871 | 27.73 | .7414 | 26.63 | .8021 | 31.24 | .9169 |
| DRCT$_1$ + $i$-LN | ×4 | **32.57** | **.8997** | **28.91** | **.7887** | **27.76** | **.7426** | **26.79** | **.8063** | **31.41** | **.9188** |

(a) Single image super-resolution (SR)

| Backbone | Testset | Metric PSNR | Metric SSIM |
|---|---|---|---|
| HAT$_1$ + LN | Rain100L | 34.35 | .9471 |
| HAT$_1$ + $i$-LN | Rain100L | **36.20** | **.9641** |
| SwinIR$_1$ + LN | Rain100L | 33.00 | .9434 |
| SwinIR$_1$ + $i$-LN | Rain100L | **34.43** | **.9527** |
| HAT$_1$ + LN | Test100 | 29.52 | .8905 |
| HAT$_1$ + $i$-LN | Test100 | **30.14** | **.9022** |
| SwinIR$_1$ + LN | Test100 | 27.45 | .8766 |
| SwinIR$_1$ + $i$-LN | Test100 | **29.87** | **.8982** |

(b) Image deraining (DR)

| Backbone | $\sigma$ | Urban100 PSNR | CBSD68 PSNR | Kodak24 PSNR | McMaster PSNR |
|---|---|---|---|---|---|
| HAT$_1$ + LN | 15 | 35.489 | 34.285 | 35.347 | 35.440 |
| HAT$_1$ + $i$-LN | 15 | **35.558** | **34.296** | **35.366** | **35.477** |
| SwinIR$_1$ + LN | 15 | 35.077 | 34.164 | 35.147 | 35.183 |
| SwinIR$_1$ + $i$-LN | 15 | **35.138** | **34.181** | **35.177** | **35.223** |
| HAT$_1$ + LN | 25 | 33.296 | 31.622 | 32.864 | 33.105 |
| HAT$_1$ + $i$-LN | 25 | **33.384** | **31.632** | **32.887** | **33.139** |
| SwinIR$_1$ + LN | 25 | 32.753 | 31.480 | 32.643 | 32.829 |
| SwinIR$_1$ + $i$-LN | 25 | **32.803** | **31.489** | **32.660** | **32.848** |

(c) Color image denoising (DN)

| Backbone | q | Urban100 PSNR | Urban100 SSIM | LIVE1 PSNR | LIVE1 SSIM | Classic5 PSNR | Classic5 SSIM |
|---|---|---|---|---|---|---|---|
| HAT$_1$ + LN | 10 | 28.45 | .8514 | 27.89 | .8048 | 29.94 | .8167 |
| HAT$_1$ + $i$-LN | 10 | **28.52** | **.8530** | **27.90** | **.8057** | **29.96** | **.8178** |
| SwinIR$_1$ + LN | 10 | 27.86 | .8400 | **27.65** | **.7995** | 29.72 | .8111 |
| SwinIR$_1$ + $i$-LN | 10 | **27.92** | **.8410** | 27.62 | .7993 | 29.72 | .8111 |
| HAT$_1$ + LN | 40 | 33.26 | .9302 | 32.63 | .9158 | 34.34 | .9060 |
| HAT$_1$ + $i$-LN | 40 | **33.36** | **.9312** | **32.67** | **.9162** | **34.39** | **.9066** |
| SwinIR$_1$ + LN | 40 | 32.62 | .9245 | 32.34 | .9127 | 34.11 | .9036 |
| SwinIR$_1$ + $i$-LN | 40 | **32.68** | **.9252** | **32.35** | **.9129** | **34.12** | **.9038** |

(d) Image JPEG compression artifact removal (CAR)

to resolve the feature increasing issue in IR tasks. Notably, these methods also align with prior studies (Lim et al., 2017b; Wang et al., 2018), where multiplying a small scale factor to the residual path components helped the network to converge. Overall, this study aims to explore 1) the feature divergence tendency of per-token and holistic normalizations and 2) determine which normalization method yields the best performance.

**Feature Divergence Behavior.** Fig.4 illustrates that feature divergence always emerges when using per-token normalizations: vanilla LN, RMSNorm, and LayerScale. In contrast, spatially consistent normalizations as our $i$-LN or BN, IN, ReZero do not exhibit the divergence trend. For the configuration without any normalization, we observe failure to converge due to unstable training. However, the feature magnitudes are well-bounded before this failure occurs, aligning with other normalization schemes without the per-token operation. This observation also aligns with our hypothesis that the feature divergence phenomena is closely related to the per-token operation, and also reveals that any spatially holistic normalization could potentially reduce this effect.

**Performance Comparisons.** We further analyze the empirical performance for each normalization scheme in Tab.1, where vanilla LN performs worst since it neglects inter-token relationships and maps features into a unified normalized space, neglecting input-dependent low-level statistics.

Per-token variants (LS, RMS) improve over vanilla LN, but underperform methods that use spatially consistent normalization. Without normalization (None), the network fails to converge, likely due to unstable gradients caused by the absence of normalization. RZ, which also omits normalization, similarly yields limited performance.

The spatially holistic variants (IN, BN) outperform those with per-token schemes (LN, LS, RMS) or normalization omitted schemes (None, RZ). However, BN leads to a significant performance drop in eval-mode, despite being healthy in train-mode; indicating the necessity of per-image statistic-based normalization in IR tasks. Meanwhile, IN performs better than vanilla LN but worse than ours. This is since IN (and also BN) discard crucial channel-wise information necessary for representing deep features, resulting in limited performance.

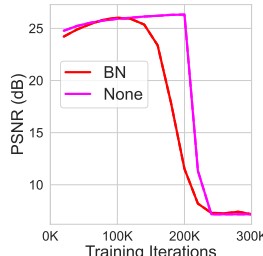

Figure 5: Eval-mode BN and removing all normalization (None) fails.

Overall, our $i$-LN achieves the best performance among all examined methods, demonstrating its effectiveness in preserving important inter-token spatial relationships and internal statistics, and ultimately the input low-level features throughout the network.

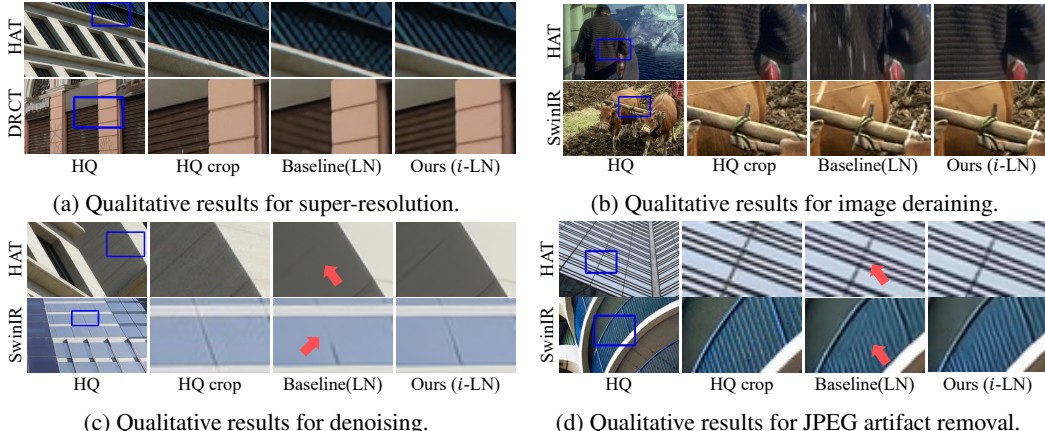

(a) Qualitative results for super-resolution.   (b) Qualitative results for image deraining.

(c) Qualitative results for denoising.   (d) Qualitative results for JPEG artifact removal.

Figure 6: Qualitative comparison across four representative image restoration tasks.

## 4.2 Analysis Under Task Variation

**Feature Divergence Behavior.** Fig. 2c illustrates the evolution of feature magnitudes across various Image Restoration (IR) tasks, including Image Super-Resolution (SR), Image Denoising (DN), Image Deraining (DR), and JPEG Compression Artifact Removal (CAR). The figure clearly demonstrates that feature divergence consistently occurs across all restoration tasks under conventional LayerNorm. In contrast, integrating our $i$-LN effectively resolves this issue, maintaining stable and well-bounded feature scales throughout training. This consistent stabilization of internal feature magnitudes confirms the general applicability and robustness of our proposed method across diverse IR scenarios.

**Benchmark: Image Super-Resolution (SR).** Tab.2a and Fig.6a illustrate quantitative and qualitative results for SR. Compared to vanilla LayerNorm, we achieve significant improvements across benchmarks. Notably, SR benefits greatly from our method due to the inherent nature of SR: the input is entirely reliable, since it exactly aligns with the low-frequency information in the ground truth. By precisely preserving these input features, our method substantially enhances restored image quality. We additionally provide a comparison against the official public models under computationally extensive settings in Appendix.B.1.

**Benchmark: Image Deraining (DR).** Similarly, Tab. 2b and Fig. 6b demonstrate substantial improvements of our method in image deraining compared to conventional LayerNorm. This improvement is particularly pronounced because our method effectively preserves reliable input regions, specifically the local areas unaffected by rain streaks. By explicitly maintaining these local correspondences with the ground truth, our $i$-LN method achieves improved restoration accuracy.

**Benchmark: Image Denoising (DN)** Tab. 2c and Fig. 6c demonstrate that our method consistently outperforms conventional LayerNorm in image denoising tasks. However, the observed performance improvements are smaller compared to SR and Deraining. This relatively reduced benefit arises because denoising involves uniformly distributed corruptions across the entire image, limiting the advantage gained from explicitly preserving particular input features. Despite this, visual examples confirm meaningful improvements in recovering sharp edges.

**Benchmark: JPEG compression artifact removal (CAR).** Similarly, Tab. 2d and Fig. 6d demonstrate consistent improvements of our method over LayerNorm for JPEG compression artifact removal. However, these performance gains remain sma ller than those achieved in SR and Deraining. Similar to denoising, JPEG artifacts affect images globally and irregularly, limiting the advantage of explicitly preserving specific input details. Still, visual examples illustrate consistent improvement in accurate artifact reductions, highlighting our method's broad effectiveness across various IR tasks.

**Benchmark: Real-world Image Restoration**. To further validate the effectiveness and robustness of the proposed method, we conduct further experiments under the challenging real-world degradation configurations. Here, we choose the representative Real-ESRGAN Wang et al. (2021) degradation pipeline and synthesize both the train and test images accordingly. Experiments are performed under the $\times 4$ SR task with the HAT$_1$ model. In Fig. 9 and Tab. 15, we provide qualitative and quantitative results, respectively. As demonstrated, our $i$-LN shows significant improvements even under the complex real-world degradation settings, successfully reconstructing fine-details and sharp edges.

## 4.3 ABLATION STUDY

To analyze the contribution of each component in $i$-LN, we conduct an ablation study by selectively removing spatial holisticness and rescaling. Compared to Tab.2, we increase the network capacity and training iterations (denoted as HAT$_2$) to ensure that the observed benefits are not simply due to faster convergence. In Tab. 3, Fig.13 and Fig.12, we provide a quantitative and qualitative analysis results under the $\times 4$ SR task. Removing either the rescaling strategy (Rs) or the spatial holisticness (SH) consistently reduces restoration quality, confirming their complementary roles in improving IR performance.

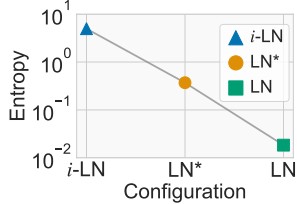

Figure 7: Channel entropy collapses *exponentially* as we remove each component (spatially holisticness and rescaling) of our $i$-LN; falling back to vanilla LN.

We then examine feature statistics in terms of channel entropy (Fig. 7). Starting from our full method, we remove the rescaling method (falling back to LN*) and subsequently also remove the spatial holistic scheme (falling back to vanilla per-token LN). Here, we observe that channel entropy collapses *exponentially*, indicating that each component contributes to maintaining well-distributed activations across channels.

Overall, using both components together achieves the best results in terms of both restoration quality and stable feature statistics. Spatial holisticness (LN*) effectively preserves inter-token relationships, while the rescaling strategy further restores the missing global scale that LN* alone cannot maintain.

## 4.4 INTRIGUING PROPERTIES UNDER VANILLA LN

### 4.4.1 HOW NETWORK SCALE IMPACTS FEATURE DIVERGENCE

We further investigate how the overall network size affects feature divergence by varying the depth and width of the IR Transformer individually. As shown in Fig. 2a–2b, larger models consistently diverge faster and to higher magnitudes. In particular, the emergence of an extreme valued feature appears to be a cumulative process: in order for a newly generated outlier channel to dominate the statistics, it must surpass the already abnormal activations propagated through the residual path, resulting in increasingly extreme values as the network scales. Taken together, our analysis reveals a potential vulnerability unique to low-level restoration at scale, where enlarging capacity does not merely amplify representational power but also exacerbates pathological feature growth.

### 4.4.2 CHANNEL IMBALANCE AND BIAS ALIGNMENT

Earlier, we observed extreme feature norms and imbalances in channel entropy, indicating highly peaky feature distributions concentrated in specific channels. Interestingly, despite these severe imbalances, baseline IR Transformers manage to converge and produce outputs with well-bounded magnitudes. To gain insights to this paradox, we take a closer look at the final normalization layer (LN). In Fig. 8, we visualize the unnormalized feature magnitudes along the channel dimension before the final normalization layer and compare them with the learned affine bias parameter ($\beta$) across various IR tasks.

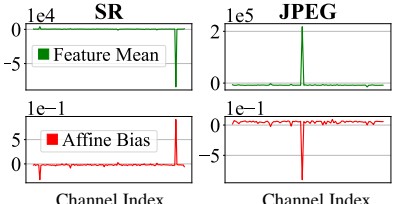

Figure 8: Alignment of affine bias parameters in the last LN and channel-wise magnitude of input feature; showing a compensatory mechanism.

We observe sharp peaks in the bias parameters precisely aligning with the channels exhibiting high magnitudes. This exact alignment reveals a compensatory mechanism where the learnable affine terms ($\gamma, \beta$) of LayerNorm counteract abnormal channel activations, allowing baselines to yield normal images. Additionally, this also indicates that the normalization operation ($\mu, \sigma$) itself is incapable of directly removing these extreme peaks.

Moreover, although the observed bias–feature alignment allows baseline IR Transformers to maintain reasonable outputs, this mechanism should be regarded as a compensatory shortcut rather than a fundamental fix. The fact that networks must rely on such peaky biases to counteract extreme channel activations leaves the model fragile and prone to failures, including potential training instability and failure in practical scenarios such as reduced-precision inference as discussed in Sec.4.5.2.

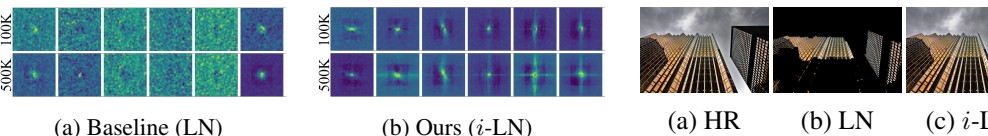

| (a) Baseline (LN) | (b) Ours ($i$-LN) |
| --- | --- |

| (a) HR | (b) LN | (c) $i$-LN |
| --- | --- | --- |

Figure 9: Visualization of Relative Position Embeddings (RPE) per head, for training iteration 100K and 500K. Ours exhibit well-structured RPEs, indicating the superiority in understanding the spatial relationship between pixels.

Figure 10: Half-precision inference results for $\times 4$ SR. LN leads to artifacts while our $i$-LN achieves near-zero fidelity loss compared to full-precision.

| Idx | Backbone | SH | Rs | BSD100 | Urban100 | Manga109 |
| --- | --- | --- | --- | --- | --- | --- |
| 1 | HAT$_2$ (LN) | | | 27.7897 | 26.8779 | 31.5444 |
| 2 | HAT$_2$ | | ✓ | 27.8615 | 27.3373 | 31.8888 |
| 3 | HAT$_2$ | ✓ | | 27.9034 | 27.5335 | 32.0837 |
| 4 | HAT$_2$ ($i$-LN) | ✓ | ✓ | **27.9206** | **27.5849** | **32.1694** |

Table 3: Ablation study. *SH* indicates introducing spatial holisticness (identical to LN*) and *Rs* indicates our rescaling strategy. Idx 1 and 4 are each identical to vanilla LN and our $i$-LN, respectively. Experiments conducted for $\times 4$ SR.

| Idx | Backbone | Quantization | | Urban100 | Manga109 |
| --- | --- | --- | --- | --- | --- |
| 1 | HAT$_2$ + LN | W | int8 | 26.8711 | 31.5266 |
| 2 | HAT$_2$ + $i$-LN | W | int8 | 27.5818 | 32.1657 |
| 3 | HAT$_2$ + LN | W | int4 | 25.0242 | 28.0831 |
| 4 | HAT$_2$ + $i$-LN | W | int4 | 26.8292 | 30.6596 |
| 5 | HAT$_2$ + LN | W+F | fp16 | 7.4640 | 5.0736 |
| 6 | HAT$_2$ + $i$-LN | W+F | fp16 | 27.5849 | 32.1693 |

Table 4: Quantitative results under low-precision inference. *W* indicates weight-only quantization, *W+F* indicates weight and feature quantization.

### 4.5 INTRIGUING PROPERTIES UNDER $i$-LN

#### 4.5.1 EVIDENCE OF ENHANCED SPATIAL CORRELATION VIA STRUCTURED RPE

Relative Position Embeddings (RPE) explicitly encodes relative spatial positions between tokens in an input-agnostic manner, similar to the convolution operation that inherently captures the spatial locality through their structured kernel patterns. Accordingly, we can consider well-structured RPEs as a strong indicator of enhanced spatial correlation understanding. In Fig.9, we analyze how our proposed normalization method influences spatial relationship modeling by visualizing the learned RPE of both the baseline IR Transformer and our proposed method. The baseline Transformer exhibits noisy, unstructured embedding patterns, suggesting a limited capability to effectively model spatial correlations. Conversely, our method produces RPEs that resemble well-structured convolutional filter patterns, clearly indicating superior capture of spatial relationships. This structured embedding aligns with our hypothesis that our spatially holistic normalization better preserves intrinsic spatial correlations, helping the network to learn spatial relations more effectively. In Fig. 12, we provide further visual examples of RPEs between LN, $i$-LN and also the ablated variants LN* and LN+Rescaling, which shows aligning results with the discussion above.

#### 4.5.2 ROBUSTNESS IN LOW PRECISION INFERENCE

Image restoration networks often require deployment on lightweight edge devices, creating significant demand for efficient inference in IR Transformers. A common approach to enhance inference efficiency is reducing precision during model deployment. Consequently, we conducted experiments under reduced-precision inference conditions to empirically evaluate the effects of $i$-LN. Initially, we applied linear weight quantization to the model weights. As shown in Tab.4, vanilla LayerNorm resulted in substantial performance degradation, while $i$-LN demonstrated remarkable stability. We further conducted half-precision inference experiments, casting both internal feature values and weights to half-precision floating-point numbers. Fig.10 illustrates that vanilla LayerNorm generated extensive regions of black dots, indicating network-generated infinity values due to extreme internal feature magnitudes inadequate for low-precision conditions. Notably, no substantial performance degradation was observed in regions where the network maintained finite feature values. This highlights the necessity of well-bounded feature values achieved by $i$-LN, emphasizing its critical role in enabling efficient inference for IR Transformers.

#### 4.5.3 EMPIRICAL EFFECTS OF IMPROVED STABILITY

To further probe training stability, we conduct extensive multi-run experiments across diverse random seeds (Fig.11). Here, we use HAT$_1$ with a smaller batch size of 2. Vanilla LN exhibits inconsistent optimization trajectories, with large fluctuations in feature statistic evolution patterns and substantial discrepancy in final PSNR results for each run. In contrast, our $i$-LN produces significantly lower variance between multiple runs, in terms of both training statistics and the final reconstruction

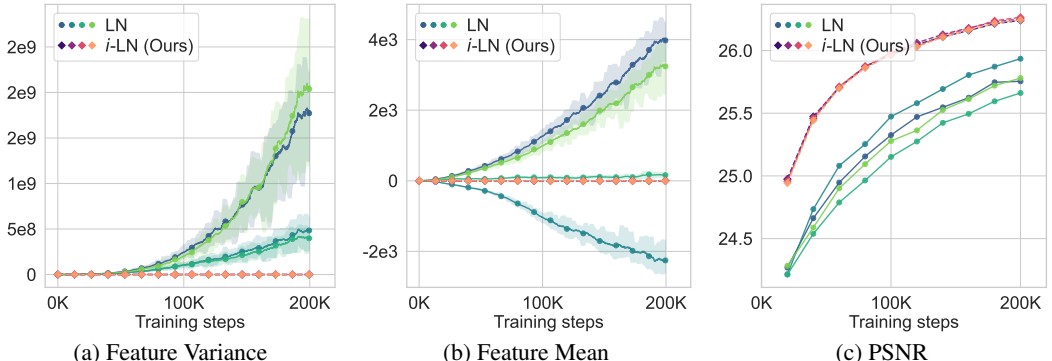

(a) Feature Variance  (b) Feature Mean  (c) PSNR

Figure 11: **Feature statistics and PSNR across multiple runs with different random seeds.** Trajectories show significant fluctuation under vanilla LN, while our $i$-LN maintains well-bounded and consistent results across all seeds ($\times 4$ SR, Urban100).

performance. These results demonstrate that $i$-LN provides a more reliable optimization landscape, reducing susceptibility to randomness in initialization or data ordering, which is an important practical advantage for training IR networks. Also refer to Appendix.E.1 for analysis over multiple batch sizes.

## 5 RELATED WORK

**Image Restoration Transformers.** Recent advances in Image Restoration (IR) transformers (Chen et al., 2024; Zamir et al., 2022; Wang et al., 2022c; Zhang et al., 2022) show superior performance over CNNs (Dong et al., 2015; Kim et al., 2016; Zhang et al., 2018) by leveraging attention mechanisms to effectively model long-range context. A pioneering work, SwinIR (Liang et al., 2021), adopted an efficient Swin-Transformer (Liu et al., 2021) based architecture in IR tasks, balancing computational cost and restoration quality. A notable method is HAT, which originated as a super-resolution model (Chen et al., 2023b) but expanded to general image restoration tasks (Chen et al., 2023a). By unifying spatial and channel attention within a hybrid attention framework, HAT surpasses existing IR Transformers in both restoration fidelity and robustness across various IR tasks.

**Abnormal Feature Behaviors.** Normalization is a key element in enhancing stability and performance in deep networks, but also can lead to unintended feature behavior. EDSR (Lim et al., 2017b), which is a foundational work in super-resolution pointed out that BatchNorm removes range flexibility of intermediate features, leading to a performance drop. Accordingly, normalization layers are removed in the most recent CNN-based SR architectures. Meanwhile, ESRGAN (Wang et al., 2018) and StyleGAN2 (Karras et al., 2020) observe that InstanceNorm and BatchNorm, respectively, cause water droplet-like artifacts. They suggest that the generator might learn to deceive feature statistics by sneaking abnormal values in internal features to reduce the effects of normalization. EDM2 (Karras et al., 2024) identifies feature magnitude divergence in diffusion models. Accordingly, they redesign the network architecture to preserve the magnitude based on statistical assumptions, leading to overall performance enhancement. DRCT (Hsu et al., 2024) notes that feature map intensities drop sharply at the end of SR networks, leading to information bottlenecks, and shows that dense residuals help.

## 6 CONCLUSION

We analyzed the training dynamics of Image Restoration (IR) Transformers and highlighted an overlooked phenomenon: divergence of feature magnitudes accompanied by collapses in channel-wise entropy. We interpret this as networks attempting to bypass the constraints of conventional LayerNorm, whose per-token normalization and input-independent scaling disrupt spatial correlations and restrict the flexibility needed for accurate restoration. To address this, we introduced Image Restoration Transformer Tailored Layer Normalization ($i$-LN), a simple drop-in replacement for LayerNorm. It is designed to better align with the unique characteristics of IR tasks and preserve important low-level features of the input throughout the network. $i$-LN normalizes jointly across spatial and channel dimensions and incorporates input-dependent rescaling, aligning normalization more closely with the demands of IR tasks. Extensive experiments show that $i$-LN prevents feature divergence, stabilizes channel entropy, improves robustness under low-precision inference, and significantly enhances IR performance across diverse tasks.

ACKNOWLEDGEMENTS

This work was supported in part by MSIT/IITP (No. RS-2022-II220680, RS-2020-II201821, RS-2019-II190421, RS-2024-00459618, RS-2024-00360227, RS-2024-00437633, RS-2024-00437102, RS-2025-25442569), MSIT/NRF (No. RS-2024-00357729), and KNPA/KIPoT (No. RS-2025-25393280).

REPRODUCIBILITY STATEMENT

Experimental settings for both training and evaluation are described in Sec.4. Detailed hyperparameter settings and network configurations for each model variant are described in Appendix.B.1 and Tab.5. Detailed algorithm to calculate the channel entropy is in Appendix.A.2 We plan to release the code for further reproducibility.

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

Table 5: Overview of model variants. ▶ indicates the group (type) of each model variant: from lightweight to computationally extensive settings. The according placements of the experiments are specified as *Main* (i.e., the main article) and *Appendix*. The placements of the detailed network hyperparameters and training configurations for each variant are highlighted in **bold**.

| Variant | Description |
|---|---|
| **▶ Type1 (Lightweight Computational Configuration)** | |
| These configurations are used for most analyses. All models were trained from scratch without the Warm-start strategy (i.e., the $\times 4$ SR models are not finetuned from the $\times 2$ SR weights), Mixing Augmentations, Progressive Patch Sizing. | |
| **SwinIR$_1$** - *Main* . . . . . . . . . . . . | Details are provided in **Tab. 10**. This variant shares the same network architecture as the official SwinIR-light model implementation. |
| **DRCT$_1$** - *Main* . . . . . . . . . . . . . | Details are provided in **Tab. 11**. This variant is a lightweight variant of the DRCT Hsu et al. (2024) model implementation. The embedding dimension is reduced in order align the network architecture with the HAT$_1$ model. |
| **HAT$_1$** - *Main & Appendix* . . . . | Details are provided in **Tab. 13**. This is a variant is a lighter version of the HAT-S Chen et al. (2023b) model, modified with a slightly reduced embedding dimension. This change was made since the standard HAT-S, despite being denoted as *small*, requires more Mult-Adds than the full-sized SwinIR model. |
| **SRFormer$_1$** - *Appendix* . . . . . . | Details are provided in **Tab. 12**. This variant is a lightweight variant of the SRFormer Zhou et al. (2023) model. The overall capacity is reduced to align with the networks specified above. |
| **▶ Type2 (Moderate Computational Configuration)** | |
| These configurations are used for the ablation study and low-precision inference analysis. | |
| **HAT$_2$** - *Main* . . . . . . . . . . . . . . | Details are provided in **Tab. 14**. This variant shares the same network capacity as the official HAT-S implementation, which is slightly heavier than HAT$_1$. However, the training budget is reduced for computational efficiency compared to HAT-S$^\dagger$ (the public model); the patch size and the batch size were halved each. Aligning with Type1 configurations, all models were trained from scratch without the warm-start strategy for 300K. |
| **▶ Type3 (Extensive Computational Configuration)** | |
| These configurations are used when comparing with official public models and validating the scalability of our method. | |
| **HAT$^\dagger$** - *Appendix* . . . . . . . . . . . . | Details are provided in **Tab. 15**. This variant shares the same network architecture as the official full-sized HAT implementation and precisely follows the training configuration of the public model. Quantitative results from this variant are copied from the original paper Chen et al. (2023b). |

Table 6: Quantitative results for classical image super-resolution under **computationally extensive setting**. † indicates that we have precisely followed the architecture and training settings of the **official public model**, as specified in Tab.15. HAT† requires 40 GPU days for ×2 SR (500K train iterations) and additional 20 GPU days for ×4 SR (250K finetuning iterations). The best results for each setting are highlighted in **bold**, respectively.

| Backbone | Scale | Set5 | | Set14 | | BSD100 | | Urban100 | | Manga109 | |
|---|---|---|---|---|---|---|---|---|---|---|---|
| | | PSNR | SSIM | PSNR | SSIM | PSNR | SSIM | PSNR | SSIM | PSNR | SSIM |
| HAT† + LN | ×2 | 38.63 | .9630 | 34.86 | .9274 | 32.62 | .9053 | 34.45 | .9466 | 40.26 | .9809 |
| HAT† + $i$-LN (Ours) | ×2 | **38.65** | **.9631** | **34.92** | **.8276** | **32.63** | **.9053** | **34.60** | **.9476** | **40.38** | **.9811** |
| HAT† + LN | ×4 | 33.04 | .9056 | 29.23 | .7973 | **28.00** | .7517 | 27.97 | .8368 | 32.48 | .9292 |
| HAT† + $i$-LN (Ours) | ×4 | **33.12** | **.9064** | **29.26** | **.7981** | **28.00** | **.7520** | **28.04** | **.8388** | **32.56** | **.9299** |

## A  EXPERIMENTAL DETAILS

### A.1  MODEL IMPLEMENTATION DETAILS

Since this work provides extensive analysis for more than 60 configurations, analyses throughout this work are performed on various settings due to computational efficiency. We provide implementation details in terms of both network architectural hyperparameters and training configuration for each model variant in Tab.5.

- **Type1 (Lightweight Setting)**: These models are the lightweight variants of the original implementations. These configurations are used in Tab.1 and Tab.2, where effects of different normalization schemes and task variations are analyzed.

- **Type2 (Moderate Setting)**: These model variants indicate moderate computational budget settings. They are used for the ablation study and also for the analysis in low-precision settings (Tab.3, Tab.4, Fig.10).

- **Type3 (Computationally Extensive Setting)**: These model variants indicate computationally extensive settings. This configuration is used to validate the scalability of our method (Tab.6), which aligns with the official implementation of the public models.

### A.2  CHANNEL ENTROPY

Algorithm 1 represents a simple pseudocode to calculate the channel-axis entropy used in our analysis. A sharp drop in channel-axis entropy indicates that feature activations are becoming concentrated in a few specific channels. Analysis throughout this work shows that this entropy collapse is intrinsically linked to the feature divergence problem that arises from conventional LayerNorm in Image Restoration (IR) Transformers.

---

**Algorithm 1** Channel Entropy Calculation

---

**Require:** Activation tensor $x$ of shape (C, H, W), a small constant $\epsilon$ for numerical stability.
**Ensure:** A single scalar entropy value.
   ▶ Step 1: Average the total activation magnitude over spatial-dim.
   ▶ Step 2: Convert to a probability distribution.
   ▶ Step 3: Compute channel entropy.

1: **function** CHANNELENTROPY($x, \epsilon$)
2:    $x_{\text{avg}} \leftarrow \text{mean}(\text{abs}(x), \text{dims} = (H, W))$               ▷ Step 1
3:    $p \leftarrow \text{softmax}(x_{\text{avg}})$                             ▷ Step 2
4:    entropy $\leftarrow$ -1 · sum($p$ · log($p + \epsilon$))        ▷ Step 3
5:    **return** entropy
6: **end function**

---

Table 7: Quantitative results for $\times 4$ super-resolution on the SRFormer (Zhou et al., 2023) network architecture. The network capacity and the training budget are adjusted as in Tab.12, which aligns with experimental settings in Tab.2. The best results for each setting are highlighted in **bold**, respectively.

| Backbone | Scale | Set5 | | Set14 | | BSD100 | | Urban100 | | Manga109 | |
|---|---|---|---|---|---|---|---|---|---|---|---|
| | | PSNR | SSIM | PSNR | SSIM | PSNR | SSIM | PSNR | SSIM | PSNR | SSIM |
| SRFormer$_1$ + LN | $\times 4$ | 32.41 | .8972 | 28.77 | .7853 | 27.68 | .7398 | 26.43 | .7957 | 30.98 | .9141 |
| SRFormer$_1$ + $i$-LN (Ours) | $\times 4$ | **32.45** | **.8979** | **28.81** | **.7862** | **27.70** | **.7407** | **26.49** | **.7997** | **31.10** | **.9152** |

Table 8: Quantitative results for $\times 4$ super-resolution with additional regularization methods. *GC* denotes Gradient Clipping, and *KLD* denotes an auxiliary KL-Divergence loss. Neither proved effective at addressing the instability caused by LayerNorm. GC slightly improves stability but still allows extreme feature magnitudes ($5.6 \times 10^6$), comparable to the vanilla baseline ($5.8 \times 10^6$). KLD regularization enforces smoother statistics but leads to a notable performance drop. In contrast, our proposed $i$-LN yields magnitudes close to $\mathcal{N}(0, 1)$ (around 1.2) while consistently outperforming all alternatives. The best results for each setting are highlighted in **bold**.

| Backbone | Scale | Set5 | | Set14 | | BSD100 | | Urban100 | | Manga109 | |
|---|---|---|---|---|---|---|---|---|---|---|---|
| | | PSNR | SSIM | PSNR | SSIM | PSNR | SSIM | PSNR | SSIM | PSNR | SSIM |
| HAT$_1$ + LN | $\times 4$ | 32.51 | .8992 | 28.79 | .7876 | 27.68 | .7411 | 26.55 | .8015 | 31.01 | .9150 |
| HAT$_1$ + LN + *GC* | $\times 4$ | 32.55 | .8996 | 28.87 | .7882 | 27.74 | .7417 | 26.70 | .8037 | 31.31 | .9169 |
| HAT$_1$ + LN + *KLD* | $\times 4$ | 32.36 | .8974 | 28.65 | .7853 | 27.64 | .7402 | 26.34 | .7972 | 30.41 | .9105 |
| HAT$_1$ + $i$-LN (Ours) | $\times 4$ | **32.72** | **.9019** | **29.01** | **.7915** | **27.84** | **.7456** | **27.17** | **.8167** | **31.82** | **.9228** |

## B  ADDITIONAL BENCHMARK RESULTS

### B.1  SCALING MODELS AND COMAPRISON AGAINST PUBLIC MODELS

In Tab.6, we validate the scalability of the proposed $i$-LN under computationally extensive settings. Specifically, we train our models on top of the full-sized HAT architecture variant, with the exact training configurations of the public model as specified in Tab.15. The models are indicated as HAT$^\dagger$, where $\dagger$ means that we have precisely followed the exact network architecture hyperparameters and training configurations for fair comparison. HAT$^\dagger$ for $\times 2$ SR and $\times 4$ SR variants requires 40 GPU days and 20 GPU days on wall-clock time each, with NVIDIA RTX A6000s under the representative `BasicSR` (Wang et al., 2022b) framework.

**Benchmark.** In Tab.6 we validate that replacing the conventional LayerNorm with the proposed $i$-LN leads to significant performance gain also in the computationally extensive setting where the networks have significantly larger capacity Accordingly, we conclude that the proposed $i$-LN is effective in both 1) lightweight settings, as shown in our main article and 2) also in computationally extensive settings as in Tab.6, showing the scalability of our $i$-LN.

**Training Details.** Scores are from the original paper for the baselines. Here, we follow the original training scheme where $\times 4$ SR models are trained under warm-start configuration (i.e., finetuned from $\times 2$ SR model weight).

### B.2  ADAPTATION TO EFFICIENT SR NETWORK

In Tab.7, we further validate the effectiveness on top of the SRFormer Zhou et al. (2023) architecture, a representative efficient SR network. Similar to other Type1 model variants, the training configurations are adjusted. Refer to Tab.12 for the detailed configurations.

**Discussion.** SRFormer utilizes a Permuted Self-Attention (PSA) mechanism. Accordingly, features across multiple pixels are reshaped into a single feature-pixel (pixelshuffle-style). Thus, the per-token vanilla LN implicitly takes normalization parameters across multi-pixels. While the effect of permuted self-attention in the perspectives of normalization was not discussed in the original work, our work suggests insights that PSA induces (partially) spatial holisticness in normalization, a potential factor for the performance gain of SRFormer (i.e., potentially reducing the performance gap against ours). Seeking further improvements regarding the relationship between the reshaping operation and normalization may be a valuable direction for future work.

## C   Other Regularization Techniques for Training Stability

Beyond our proposed $i$-LN, one may ask whether simpler regularization methods could mitigate the training instabilities of IR Transformers. We therefore examined common strategies such as gradient clipping (GC) and KL divergence (KLD) regularization in Tab.8.

While GC is widely used to bound exploding gradients, our experiments confirmed that it does not prevent the emergence of extreme feature magnitudes in IR Transformers. The maximum feature magnitude observed during training with GC was $5.6 \times 10^6$. As a reference, the maximum feature magnitude for the vanilla $HAT_1$+LN was $5.8 \times 10^6$. In contrary, our $HAT_1$+$i$-LN shows 1.2, very closely aligning with the expected magnitude of a random noise sampled from the normal distribution $\mathcal{N}(0, 1)$, which is 1. Additionally, while GC leads to slight performance improvement against the vanilla model, it consistently underperforms compared to $i$-LN.

Likewise, KLD regularization can stabilize feature statistics, but at the cost of substantial reconstruction performance degradation. Specifically, we observed that although KLD encourages well-behaved distributions, the resulting models suffered PSNR drops even below the baseline with vanilla LN. This is consistent with prior findings in rate–distortion theory (Brekelmans et al., 2019; Blau & Michaeli, 2019), and also to VAE literature (Higgins et al., 2017; Yao et al., 2025), where strong regularization penalties reduce reconstruction fidelity.

Overall, these results highlight that although general-purpose regularization may offer partial remedies, they are either ineffective (GC) or detrimental to reconstruction quality (KLD). This further emphasizes the necessity of normalization methods tailored to the unique requirements of IR Transformers.

## D   Further Ablation Study

Here, we further compare our full method $i$-LN against the baseline (vanilla) LN, and two variants: LN* and LN+Rescaling, which each are ablations of out method without rescaling (LN*) and without the spatial holistic type of normalization (LN+Rescaling). Below, we provide additional visualization of relative position embeddings (RPEs) and also further visual comparison of the according $\times 4$ SR result. Quantitative comparison can be seen in Tab.3 of the main article. Each of these experimental setups is directly aligned with the ablation analysis presented in Tab. 3 of the main article. Accordingly, all experiments were performed with the $HAT_2$ configuration for the synthetic (bicubic) $\times 4$ SR task.

### D.1   Ablation Study: Relative Position Embeddings (RPE) Visualization

In Fig. 12, we provide a comprehensive visual comparison of the learned Relative Position Embeddings (RPE) across different normalization strategies. The figure compares vanilla Layer Normalization (LN), our proposed $i$-LN, as well as two ablated variants, LN* and LN+Rescaling.

To obtain deeper insight into the dynamics and stability of RPE formation, we visualize both early-stage training (100K iterations) and fully converged models (500K iterations), and further examine representations from a shallow block (`RHAG.0`) and a deep block (`RHAG.5`).

Across all settings, vanilla LN exhibits highly noisy RPE structures throughout the entire training process. Even after convergence, its embeddings fail to organize into meaningful spatial patterns, suggesting a limited ability to encode coherent spatial correlations. The LN* variant, which adopts a spatially holistic normalization, occasionally reveals global structures, but these patterns remain weak and are overshadowed by considerable noise. The LN+Rescaling variant shows improved structure in deeper layers, yet its shallow-layer embeddings remain unstable and inconsistent. This indicating that rescaling alone is insufficient to guide early-layer RPE formation, reflected in low reconstruciton scores in Tab.3.

In contrast, our proposed $i$-LN consistently produces substantially clearer and more structured RPE maps, with significantly reduced noise across both shallow and deep layers and across all training stages. The strong spatial coherence visible in the embeddings aligns with the quantitative improvements reported in Tab. 3, where $i$-LN achieves the highest performance among all evaluated variants. These visual results confirm that $i$-LN facilitates stable and meaningful spatial relational modeling throughout the entire network depth and training trajectory.

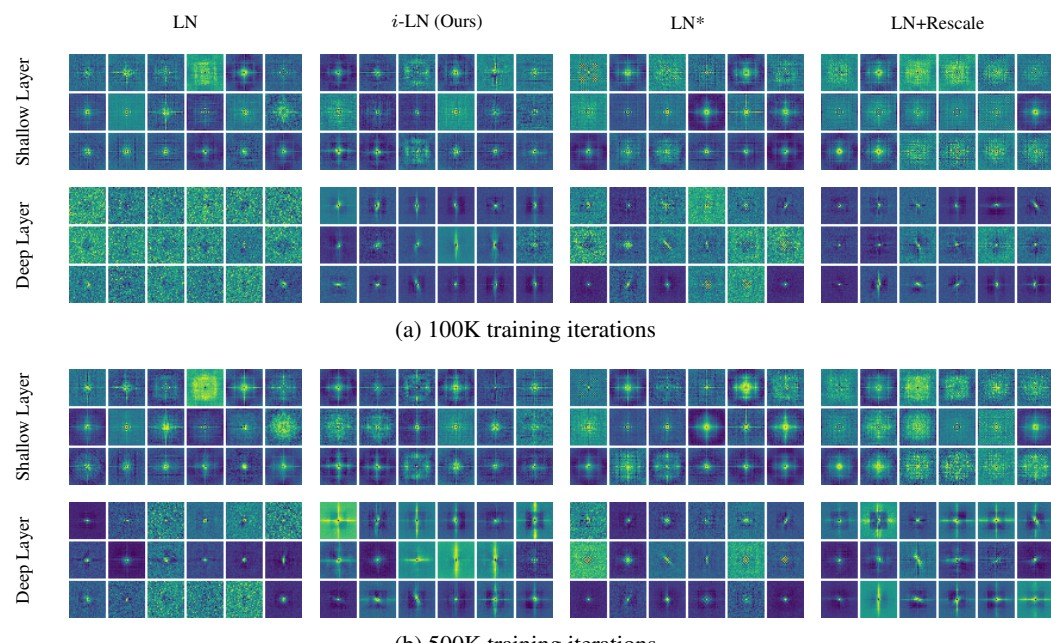

Figure 12: Visual comparison of Relative Position Embeddings (RPE) across attention heads. We show the RPEs from the last three attention layers of each a shallow (`RHAG.0`) and a deep (`RHAG.5`) building block of HAT Chen et al. (2023a) (i.e., the `RHAG` Block), as well as early training (100K iterations) versus fully converged models (500K iterations). Each corresponds to an experimental setting aligned with Tab. 3, where vanilla LN and our $i$-LN are the primary comparison, and LN* and LN+Rescale represent ablation variants obtained by selectively removing components of our method. Our full method ($i$-LN) yields cleaner and more stable RPE patterns, with substantially reduced noise across variations in training iteration and layer depth.

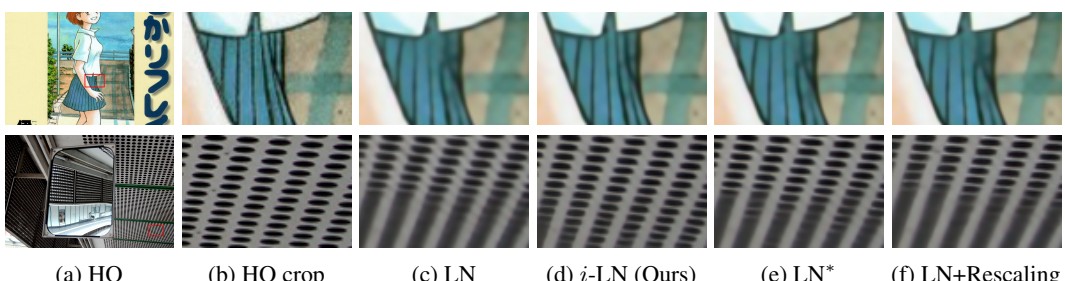

(a) HQ  (b) HQ crop  (c) LN  (d) $i$-LN (Ours)  (e) LN*  (f) LN+Rescaling

Figure 13: Visual comparison between LN and $i$-LN, along with the ablated variants LN* and LN+Rescaling with the $HAT_2$ configuration. Experimental settings follow the ablation study in the main article (Tab. 3). The proposed $i$-LN more faithfully reconstructs fine details, producing sharper edges and clearer complex patterns than the LN baseline.

## D.2 ABLATION STUDY: ×4 SR VISUALIZATION

In Fig.13, we provide additional visual comparisons between the SR outputs obtained by networks each employing LN, LN*, LN+Rescale and our i-LN.

Across all cases, the model equipped with $i$-LN produces the sharpest and most faithful reconstruction of fine-grained structures, including thin edges, repetitive patterns, and high-frequency textures. The restored images exhibit not only improved clarity but also enhanced local contrast and reduced artificial smoothing, indicating that $i$-LN effectively preserves low-level feature statistics throughout the network.

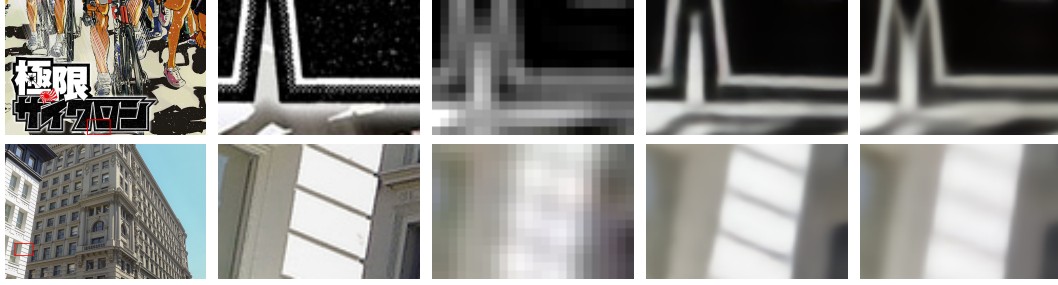

|  (a) HQ | (b) HQ crop | (c) Input | (d) $i$-LN (Ours) | (e) LN |

Figure 15: Visual comparison between LN and $i$-LN for the real-world $\times 4$ super-resolution task with $HAT_1$. Both the training images and the test images were synthesized following the Real-ESRGAN (Wang et al., 2021) degradation pipeline.

By contrast, the baseline vanilla LN often yields blurry and overly smoothed outputs, where crucial high-frequency details are lost. This degradation is consistent with the unstable and noisy RPE behavior observed earlier, suggesting that vanilla LN struggles to maintain coherent spatial relations required for accurate detail reconstruction. The ablated variants as LN* (spatial holistic normalization without rescaling) and LN+Rescaling (rescaling without spatial holisticness) show partial improvements over LN but still fall short of $i$-LN.

Overall, the qualitative comparisons provide visual evidence that both components of $i$-LN (spatial holisticness and input-adaptive rescaling) are essential.

# E  ADDITIONAL EXPERIMENTS ON THE STABILITY AND ROBUSTNESS

In this section, we provide additional experiments that further validate the stability, robustness, and general applicability of the proposed $i$-LN across a wide range of practical and challenging training scenarios. Specifically, we evaluate: 1) real-world super-resolution settings, 2) robustness under multiple training batch sizes.

Across all settings, $i$-LN consistently outperforms and exhibits more stable behavior than the conventional per-token LayerNorm (LN). Unless otherwise specified, the base configuration follows the $HAT_1$ setting for the $\times 4$ SR task. In all experiments, we compare against the vanilla LN baseline under identical training configurations.

## E.1  ROBUSTNESS ACROSS VARYING BATCH SIZES

To evaluate the robustness of $i$-LN, we train $HAT_1$ models but with varying batch size from 2 to 8; while keeping all other hyperparameters fixed (Fig. 14). This experimental design is chosen in order to mimic unstable training configurations without heavy hyperparameter search. For each configuration, we report PSNR and SSIM for the $\times 4$ SR task on Urban100 throughout training. As shown in Fig. 14, models with $i$-LN consistently achieve higher PSNR/SSIM than the baseline across all batch sizes. Notably, the performance gap remains stable as the batch size decreases. These results demonstrate that the benefit of $i$-LN is not tied to a particular training setup, and that its stability advantages persist even under extremely small-batch training (e.g., batch size 2). This property is especially valuable for memory-constrained environments where large batches are infeasible.

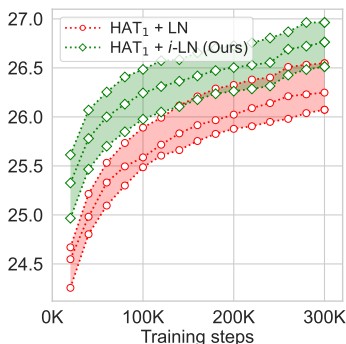

Figure 14: PSNR for $\times 4$ SR with the $HAT_1$ model, but with batch size 2, 4, 8 (Urban100).

## E.2  REAL-WORLD SUPER-RESOLUTION

To evaluate the practical effectiveness of $i$-LN, we adopt a real-world degradation setup following the RealESRGAN Wang et al. (2021) pipeline. These analyses

| Run | Backbone | Set5 | | Set14 | | BSD100 | | Urban100 | | Manga109 | |
|---|---|---|---|---|---|---|---|---|---|---|---|
| | | PSNR | SSIM | PSNR | SSIM | PSNR | SSIM | PSNR | SSIM | PSNR | SSIM |
| 1 | RealHAT$_1$ + LN | 26.27 | .7601 | 24.43 | .6272 | 24.39 | .5881 | 22.01 | .6064 | 23.65 | .7466 |
| | RealHAT$_1$ + $i$-LN | **26.38** | **.7656** | **24.58** | **.6322** | **24.47** | **.5918** | **22.19** | **.6146** | **23.96** | **.7557** |
| 2 | RealHAT$_1$ + LN | 26.77 | .7739 | 24.27 | .6201 | 24.29 | .5839 | 22.00 | .6056 | 23.61 | .7503 |
| | RealHAT$_1$ + $i$-LN | **26.90** | **.7777** | **24.46** | **.6272** | **24.37** | **.5876** | **22.19** | **.6141** | **23.84** | **.7587** |
| 3 | RealHAT$_1$ + LN | 24.39 | .6927 | 24.77 | .6325 | 24.32 | .5823 | 21.96 | .5973 | 23.71 | .7522 |
| | RealHAT$_1$ + $i$-LN | **24.58** | **.6993** | **24.94** | **.6377** | **24.40** | **.5862** | **22.16** | **.6055** | **23.99** | **.7602** |
| 4 | RealHAT$_1$ + LN | 25.86 | .7472 | 24.72 | .6389 | 24.35 | .5878 | 21.84 | .5954 | 23.22 | .7365 |
| | RealHAT$_1$ + $i$-LN | **26.05** | **.7531** | **24.93** | **.6447** | **24.43** | **.5913** | **22.04** | **.6046** | **23.48** | **.7450** |
| 5 | RealHAT$_1$ + LN | 25.08 | .7086 | 24.46 | .6355 | 24.38 | .5899 | 21.93 | .6032 | 23.46 | .7458 |
| | RealHAT$_1$ + $i$-LN | **25.26** | **.7120** | **24.63** | **.6411** | **24.46** | **.5933** | **22.10** | **.6115** | **23.73** | **.7554** |
| Avg. | RealHAT$_1$ + LN | 25.68 | .7365 | 24.53 | .6308 | 24.35 | .5864 | 21.95 | .6016 | 23.53 | .7463 |
| | RealHAT$_1$ + $i$-LN | **25.83** | **.7415** | **24.71** | **.6366** | **24.43** | **.5900** | **22.14** | **.6101** | **23.80** | **.7550** |

Table 9: Quantitative results for real-world $\times 4$ super-resolution across five random seeds. Both the training images and the test images were synthesized following the Real-ESRGAN (Wang et al., 2021) degradation pipeline. The best results for each setting are highlighted in **bold**.

complement the main text by demonstrating that the advantages of $i$-LN are not restricted to controlled laboratory settings, but generalize to real-world usage and unstable training regimes.

Training is performed on synthetic DF2K pairs, and testing is conducted on synthetically degraded versions of standard SR benchmarks (Set5, Set14, BSD100, Urban100, Manga109). Degradation synthesis strictly follows the RealESRGAN procedures. To assess robustness, we repeat each experiment across five different random seeds and report the average score of the resulting PSNR/SSIM scores.

As shown in Tab.9 and Fig.15, $i$-LN leads to consistent and significant improvements over vanilla LN across all benchmarks, despite the increased complexity of the real-world degradation pipeline. These results highlight that $i$-LN is not only theoretically grounded but also practically beneficial in more challenging real-world restoration scenarios.

## F  ADDITIONAL QUALITATIVE COMPARISON

We provide additional visual comparisons between IR Transformers with our proposed $i$-LN against their counterparts using vanilla Layer Normalization (LN). As shown in the following figures, $i$-LN consistently produces sharper structures, cleaner textures, and fewer artifacts across a range of low-level vision tasks. Qualitative results are provided for: (i) super-resolution (Figs. 16 and 17), (ii) image denoising (Figs. 18 and 19), (iii) JPEG compression artifact removal (Figs. 23 and 22), and (iv) image deraining (Figs. 20 and 21).

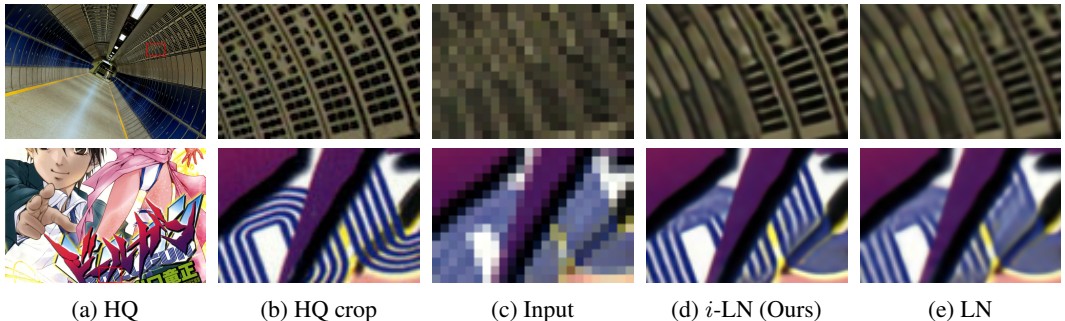

(a) HQ     (b) HQ crop     (c) Input     (d) $i$-LN (Ours)     (e) LN

Figure 16: Visual comparison between LN and $i$-LN for the super-resolution task with $HAT_1$.

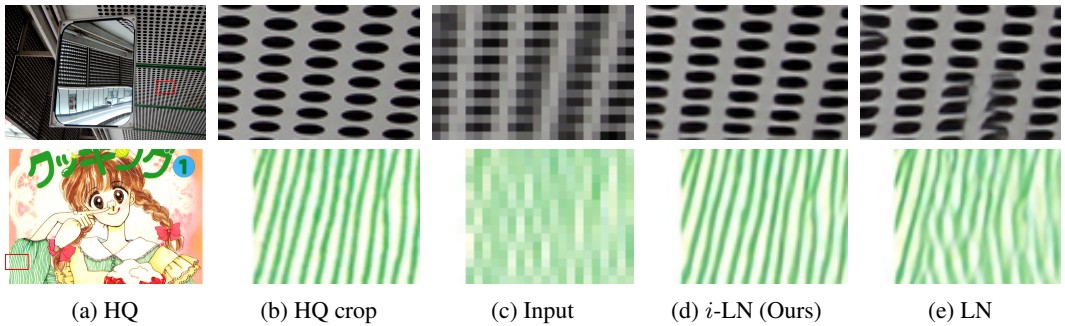

(a) HQ     (b) HQ crop     (c) Input     (d) $i$-LN (Ours)     (e) LN

Figure 17: Visual comparison between LN and $i$-LN for the super-resolution task with $DRCT_1$.

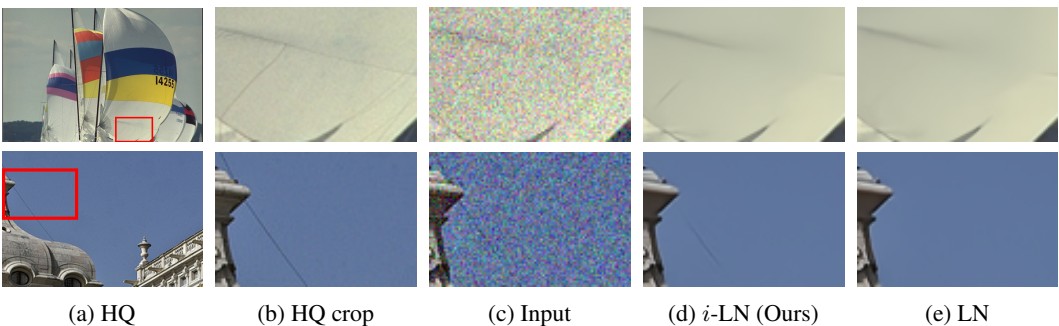

(a) HQ     (b) HQ crop     (c) Input     (d) $i$-LN (Ours)     (e) LN

Figure 18: Visual comparison between LN and $i$-LN for the denoising task with $HAT_1$.

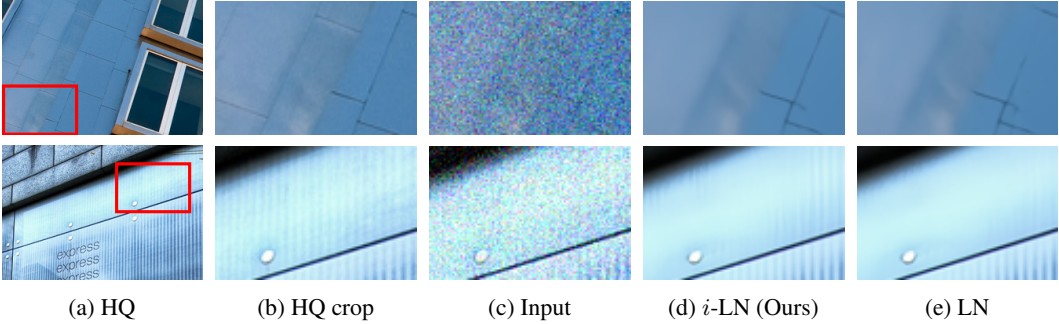

(a) HQ     (b) HQ crop     (c) Input     (d) $i$-LN (Ours)     (e) LN

Figure 19: Visual comparison between LN and $i$-LN for the denoising task with $SwinIR_1$.

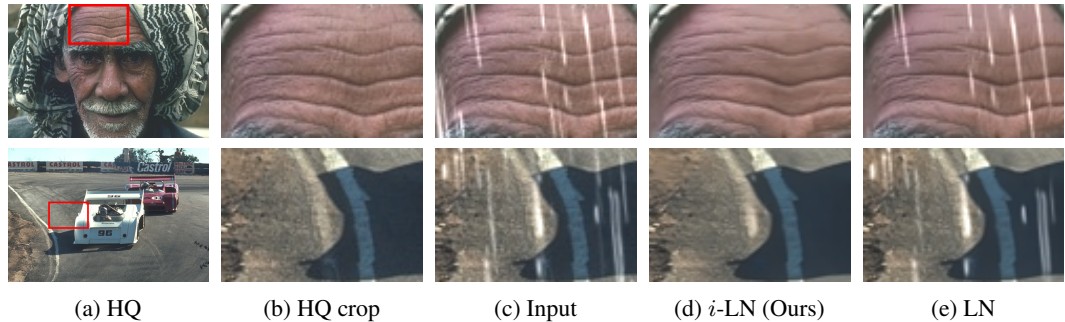

(a) HQ      (b) HQ crop      (c) Input      (d) $i$-LN (Ours)      (e) LN

Figure 20: Visual comparison between LN and $i$-LN for the deraining task with HAT$_1$.

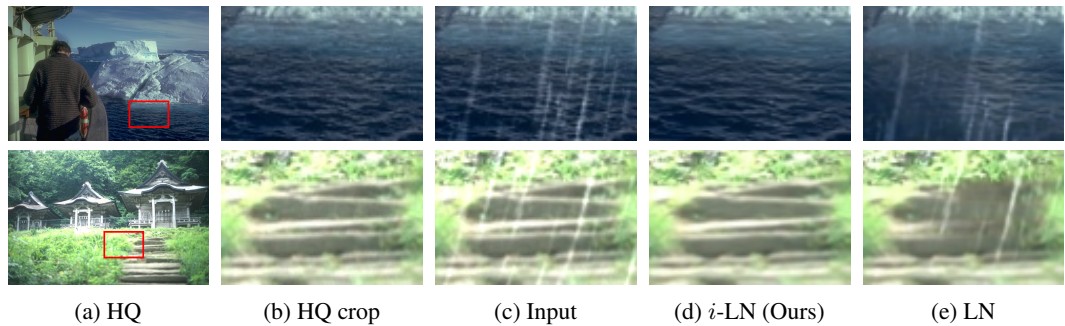

(a) HQ      (b) HQ crop      (c) Input      (d) $i$-LN (Ours)      (e) LN

Figure 21: Visual comparison between LN and $i$-LN for the deraining task with SwinIR$_1$.

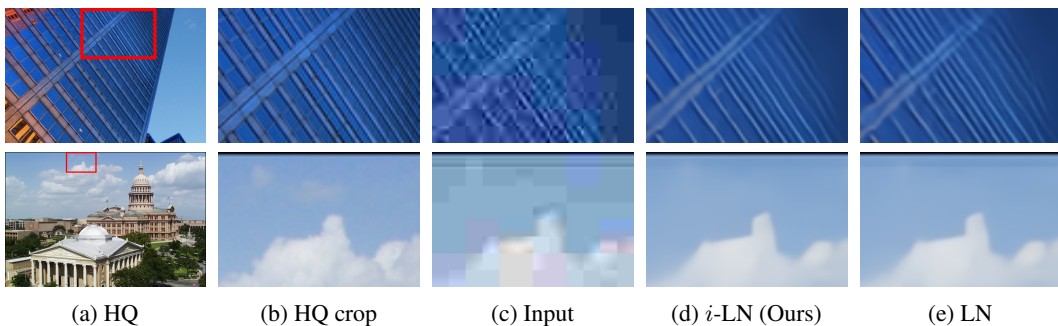

(a) HQ      (b) HQ crop      (c) Input      (d) $i$-LN (Ours)      (e) LN

Figure 22: Visual comparison between LN and $i$-LN for the JPEG artifact removal task with HAT$_1$.

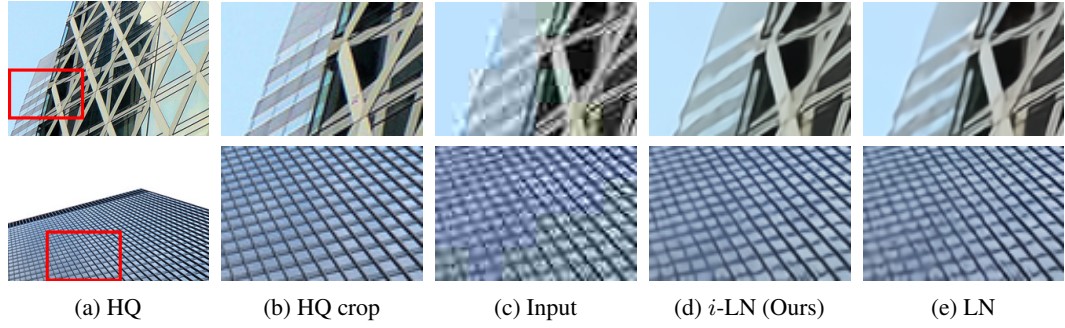

(a) HQ      (b) HQ crop      (c) Input      (d) $i$-LN (Ours)      (e) LN

Figure 23: Visual comparison between LN and $i$-LN for the JPEG artifact removal task with SwinIR$_1$.

# G    DETAILED DERIVATIONS FOR STRUCTURE PRESERVATION

## G.1    NOTATION AND PRELIMINARIES

Let $X \in \mathbb{R}^{L \times C}$ be the feature matrix with tokens $x_\ell \in \mathbb{R}^C$ (row-vectors). Define the inter-pixel (inter-token) structure by the set of pairwise displacements

$$\Delta X := \{x_\ell - x_k : 1 \le \ell, k \le L\}.$$

A map $T : \mathbb{R}^C \to \mathbb{R}^C$ *preserves inter-pixel structure up to scale* if there exists a homothety $H(x) = ax + b$ with $a > 0$, $b \in \mathbb{R}^C$ such that

$$T(x_\ell) - T(x_k) = a(x_\ell - x_k) \quad \text{for all } \ell, k.$$

Equivalently, all angles and pairwise distance ratios are preserved.

We analyze the *pure normalization maps* (i.e., the normalization before the affine $(\gamma, \beta)$ is applied); any global translation by $\beta$ does not affect $\Delta X$, and a scalar post-scale can be absorbed into the homothety factor $a$.

## G.2    PROPOSITION 1 (VANILLA LAYERNORM FAILS TO PRESERVE STRUCTURE)

Let $T_{\mathrm{LN}}$ denote the transformation defined by the normalization in vanilla *per-token* LayerNorm. In general there do not exist $a > 0$ and an orthogonal $Q$ such that for all tokens $x_\ell, x_k$,

$$T_{\mathrm{LN}}(x_\ell) - T_{\mathrm{LN}}(x_k) = a\,Q(x_\ell - x_k).$$

Hence $T_{\mathrm{LN}}$ is not conformal on the token set. By the nested class relation $\mathrm{Homothety} \subset \mathrm{Similarity} \subset \mathrm{Conformal}$, it follows that $T_{\mathrm{LN}}$ is neither a similarity nor a homothety, and thus does *not* preserve inter-pixel structure in general.

**Proof.**    Write per-token means and standard deviations as

$$\mu_\ell \;=\; \frac{1}{C} \sum_{c=1}^{C} x_{\ell,c}, \qquad \sigma_\ell \;=\; \Big( \frac{1}{C} \sum_{c=1}^{C} (x_{\ell,c} - \mu_\ell)^2 \Big)^{1/2}.$$

The pure LN map (no $\gamma, \beta$) acts componentwise as

$$T_{\mathrm{LN}}(x_\ell) \;=\; \frac{x_\ell - \mu_\ell \mathbf{1}}{\sigma_\ell},$$

so for two tokens $\ell, k$,

$$\Delta_{\ell k} := T_{\mathrm{LN}}(x_\ell) - T_{\mathrm{LN}}(x_k) \tag{4}$$

$$= \frac{x_\ell}{\sigma_\ell} - \frac{x_k}{\sigma_k} - \Big( \frac{\mu_\ell}{\sigma_\ell} - \frac{\mu_k}{\sigma_k} \Big) \mathbf{1}. \tag{1}$$

Assume for contradiction there exist $a > 0$ and orthogonal $Q$ such that $\Delta_{\ell k} = a\,Q(x_\ell - x_k)$ for *all* $\ell, k$. Compare the coefficients of $x_\ell$ and $x_k$ on both sides of (1). Because the equality must hold for arbitrary token values, we must have

$$\frac{1}{\sigma_\ell} I = aQ \quad \text{and} \quad \frac{1}{\sigma_k} I = aQ,$$

hence $\sigma_\ell = \sigma_k$ for all $\ell, k$ and $Q$ must be proportional to the identity. With $\sigma_\ell \equiv \sigma$, the bias term in (1) reduces to $\big( \frac{\mu_k - \mu_\ell}{\sigma} \big) \mathbf{1}$, which must vanish for all $\ell, k$; thus $\mu_\ell = \mu_k$ for all $\ell, k$. Therefore the assumed similarity can hold only in the *degenerate* case where all tokens share identical per-token mean and variance.

For real features, $\{\mu_\ell, \sigma_\ell\}$ are not constant across tokens, so the assumption leads to a contradiction. Hence no single similarity map exists; $T_{\mathrm{LN}}$ is not conformal and does not preserve spatial structure.

**Remark.**    The degenerate equal-statistics case is precisely the rare exception noted in the main text.

### G.3   PROPOSITION 2 (LN* PRESERVES STRUCTURE)

Let $T_{\mathrm{LN}^*}$ denote the transformation defined by the normalization in *spatially holistic* LayerNorm (LN*) with global mean and standard deviation

$$\mu = \frac{1}{LC} \sum_{\ell,c} x_{\ell,c}, \qquad \sigma = \Big( \frac{1}{LC} \sum_{\ell,c} (x_{\ell,c} - \mu)^2 \Big)^{1/2} > 0.$$

Then for any tokens $x_\ell, x_k$,

$$T_{\mathrm{LN}^*}(x_\ell) - T_{\mathrm{LN}^*}(x_k) = \frac{1}{\sigma}(x_\ell - x_k),$$

so $T_{\mathrm{LN}^*}$ is a homothety and preserves inter-pixel structure up to a global scale.

**Proof.**   $T_{\mathrm{LN}^*}$ (without $\gamma, \beta$) is

$$T_{\mathrm{LN}^*}(x) \;=\; \frac{x - \mu \mathbf{1}}{\sigma}.$$

Hence

$$T_{\mathrm{LN}^*}(x_\ell) - T_{\mathrm{LN}^*}(x_k) = \frac{x_\ell - \mu \mathbf{1}}{\sigma} - \frac{x_k - \mu \mathbf{1}}{\sigma} = \frac{1}{\sigma}(x_\ell - x_k).$$

This is exactly a homothety with scale factor $a = \sigma^{-1}$; therefore angles and pairwise distance ratios of $\Delta X$ are preserved and the spatial configuration is rigid up to a uniform scale.

**Network Architecture Hyperparameters**

| | |
|---|---|
| Embedding Dimension | 60 |
| Layer Depths | [6, 6, 6, 6] |
| Attention Heads | [6, 6, 6, 6] |
| Window Size | $8 \times 8$ |
| MLP Ratio | 2 |
| Residual Connection | '1conv' |

**Dataset Configuration**

| | |
|---|---|
| Training Dataset | DIV2K + Flickr2K |
| PatchSize \| BatchSize | |
|   - Denoising | $64 \times 64 \mid 16$ |
|   - Deraining | $128 \times 128 \mid 8$ |
|   - JPEG Artifact Removal | $64 \times 64 \mid 16$ |
| Noise Degradation | `torch.randn` |
| JPEG Degradation | `OpenCV` |

**Optimizing Configuration**

| | |
|---|---|
| Total Iterations | 300K |
| Optimizer | Adam |
| Learning Rate (LR) | $2 \times 10^{-4}$ |
| Adam Betas | (0.9, 0.99) |
| Weight Decay | 0 |
| Scheduler ($\gamma = 0.5$) | StepLR |
| Milestones (K) | 250 |
| Loss Function | L1 Loss |

Table 10: **Hyperparameters and training configurations for the model variant SwinIR$_1$.** Network architectural terminology is based on terminologies either in the official implementation of each work or the implementation in `BasicSR` Wang et al. (2022b).

**Network Architecture Hyperparameters**

| | |
|---|---|
| Embedding Dimension | 96 |
| Layer Depths | [6, 6, 6, 6, 6, 6] |
| Attention Heads | [6, 6, 6, 6, 6, 6] |
| Window Size | $16 \times 16$ |
| MLP Ratio | 2 |
| Residual Connection | '1conv' |

**Dataset Configuration**

| | |
|---|---|
| Training Dataset | DIV2K + Flickr2K |
| PatchSize \| BatchSize | |
|   - $\times 2$ Super Resolution | $64 \times 64 \mid 16$ |
|   - $\times 4$ Super Resolution | $128 \times 128 \mid 16$ |
| SR Degradation | `MATLAB` |

**Optimizing Configuration**

| | |
|---|---|
| Total Iterations | 300K |
| Optimizer | Adam |
| Learning Rate (LR) | $2 \times 10^{-4}$ |
| Adam Betas | (0.9, 0.99) |
| Weight Decay | 0 |
| Scheduler ($\gamma = 0.5$) | StepLR |
| Milestones (K) | 250 |
| Loss Function | L1 Loss |

Table 11: **Hyperparameters and training configurations for the model variant DRCT$_1$.** Network architectural terminology is based on terminologies either in the official implementation of each work or the implementation in `BasicSR` Wang et al. (2022b).

**Network Architecture Hyperparameters**

| | |
|---|---|
| Embedding Dimension | 60 |
| Layer Depths | [6, 6, 6, 6] |
| Attention Heads | [6, 6, 6, 6] |
| Window Size | $16 \times 16$ |
| MLP Ratio | 2 |
| Residual Connection | '1conv' |

**Dataset Configuration**

| | |
|---|---|
| Training Dataset | DIV2K + Flickr2K |
| PatchSize \| BatchSize | |
| - $\times 4$ Super Resolution | $128 \times 128 \mid 16$ |
| SR Degradation | MATLAB |

**Optimizing Configuration**

| | |
|---|---|
| Total Iterations | 300K |
| Optimizer | Adam |
| Learning Rate (LR) | $2 \times 10^{-4}$ |
| Adam Betas | (0.9, 0.99) |
| Weight Decay | 0 |
| Scheduler ($\gamma = 0.5$) | StepLR |
| Milestones (K) | 250 |
| Loss Function | L1 Loss |

Table 12: **Hyperparameters and training configurations for the model variant SRFormer$_1$.** Network architectural terminology is based on terminologies either in the official implementation of each work or the implementation in `BasicSR` Wang et al. (2022b).

**Network Architecture Hyperparameters**

| | |
|---|---|
| Embedding Dimension | 96 |
| Layer Depths | [6, 6, 6, 6, 6, 6] |
| Attention Heads | [6, 6, 6, 6, 6, 6] |
| Window Size | $16 \times 16$ |
| MLP Ratio | 2 |
| Compress Ratio | 24 |
| Squeeze Factor | 24 |
| Overlap Ratio | 0.5 |
| Conv Scale | 0.01 |
| Residual Connection | '1conv' |

**Dataset Configuration**

| | |
|---|---|
| Training Dataset | DIV2K + Flickr2K |
| PatchSize \| BatchSize | |
| - Denoising | $64 \times 64 \mid 16$ |
| - Deraining | $128 \times 128 \mid 8$ |
| - JPEG Artifact Removal | $64 \times 64 \mid 16$ |
| - $\times 2$ Super Resolution | $64 \times 64 \mid 16$ |
| - $\times 4$ Super Resolution | $128 \times 128 \mid 16$ |
| Noise Degradation | `torch.randn` |
| JPEG Degradation | `OpenCV` |
| SR Degradation | MATLAB |

**Optimizing Configuration**

| | |
|---|---|
| Total Iterations | 300K |
| Optimizer | Adam |
| Learning Rate (LR) | $2 \times 10^{-4}$ |
| Adam Betas | (0.9, 0.99) |
| Weight Decay | 0 |
| Scheduler ($\gamma = 0.5$) | StepLR |
| Milestones (K) | 250 |
| Loss Function | L1 Loss |

Table 13: **Hyperparameters and training configurations for the model variant HAT$_1$.** Network architectural terminology is based on terminologies either in the official implementation of each work or the implementation in `BasicSR` Wang et al. (2022b).

| Network Architecture Hyperparameters | |
|---|---|
| **Embedding Dimension** | **144** |
| Layer Depths | [6, 6, 6, 6, 6, 6] |
| Attention Heads | [6, 6, 6, 6, 6, 6] |
| Window Size | $16 \times 16$ |
| MLP Ratio | 2 |
| Compress Ratio | 24 |
| Squeeze Factor | 24 |
| Overlap Ratio | 0.5 |
| Conv Scale | 0.01 |
| Residual Connection | '1conv' |
| **Dataset Configuration** | |
| Training Dataset | DIV2K + Flickr2K |
| PatchSize \| BatchSize | |
|   - $\times 4$ Super Resolution | $128 \times 128 \mid 16$ |
| SR Degradation | `MATLAB` |
| **Optimizing Configuration** | |
| **Total Iterations** | **500K** |
| Optimizer | Adam |
| Learning Rate (LR) | $2 \times 10^{-4}$ |
| Adam Betas | (0.9, 0.99) |
| Weight Decay | 0 |
| **Scheduler ($\gamma = 0.5$)** | **MultiStepLR** |
| **Milestones (K)** | **[250, 400, 450, 475]** |
| Loss Function | L1 Loss |

Table 14: **Hyperparameters and training configurations for HAT$_2$.** This variant uses the **HAT-S** architecture but is trained with a reduced budget. Differences from HAT$_1$ are highlighted in **bold**. Network architectural terminology is based on terminologies either in the official implementation of each work or the implementation in `BasicSR` Wang et al. (2022b).

| Network Architecture Hyperparameters | |
|---|---|
| **Embedding Dimension** | 180 |
| Layer Depths | [6, 6, 6, 6, 6, 6] |
| Attention Heads | [6, 6, 6, 6, 6, 6] |
| Window Size | $16 \times 16$ |
| MLP Ratio | 2 |
| **Compress Ratio** | 3 |
| **Squeeze Factor** | 30 |
| Overlap Ratio | 0.5 |
| Conv Scale | 0.01 |
| Residual Connection | '1conv' |
| Dataset Configuration | |
| Training Dataset | DIV2K + Flickr2K |
| PatchSize \| BatchSize | |
|   - $\times 2$ Super Resolution | $128 \times 128$ \| 32 |
|   - $\times 4$ Super Resolution | $256 \times 256$ \| 32 |
| SR Degradation | `MATLAB` |
| Optimizing Configuration | |
| Optimizer | Adam |
| Adam Betas | (0.9, 0.99) |
| Weight Decay | 0 |
| Scheduler ($\gamma = 0.5$) | MultiStepLR |
| Loss Function | L1 Loss |
| $\times 2$ Super Resolution | |
|   - Total Iterations | 500K |
|   - Learning Rate (LR) | $2 \times 10^{-4}$ |
|   - Scheduler Milestones (K) | [250, 400, 450, 475] |
| $\times 4$ Super Resolution | |
|   - Total Iterations | 250K |
|   - Learning Rate (LR) | $1 \times 10^{-4}$ |
|   - Scheduler Milestones (K) | [125, 200, 225, 240] |
|   - Pretrained | finetune from $\times 2$ SR weight |

Table 15: **Hyperparameters and training configurations for HAT[†].** This variant uses the full-sized **HAT** architecture and precisely follows the training settings of the public model. Network architectural terminology is based on terminologies either in the official implementation of each work or the implementation in `BasicSR` Wang et al. (2022b).

## H  THE USE OF LARGE LANGUAGE MODELS (LLMS)

In this study, LLMs were used for text editing, grammar correction, and coding assistance for graph visualization.

