# OpenReview forum: "Analyzing the Training Dynamics of Image Restoration Transformers: A Revisit to Layer Normalization"
_ICLR.cc/2026/Conference — ICLR 2026 Poster_

### Official Review · Reviewer_U1uB · 2025-10-27

**Soundness:** 3
**Presentation:** 3
**Contribution:** 3
**Rating:** 6
**Confidence:** 5

**Summary:**

This paper analyzes the training dynamics of the image restoration Transformer, identifies the limitations of layer normalization, and proposes a global version of layer normalization that normalizes features holistically. Through theoretical analysis and experimental verification, its effectiveness in various tasks of image restoration is demonstrated.

**Strengths:**

1. The paper is well-written and the theoretical analysis is solid.
2. The method is very simple and has been verified for its effectiveness in various IR tasks. I highly appreciate its motivation of specifically designing methods for low-level visual tasks. This is more impressive compared to simply transplanting methods from other fields directly.
3.This method is plug-and-play and may become the standard approach for designing network structures in the field of low-level vision.

**Weaknesses:**

No obvious weaknesses are identified.

**Questions:**

1. Could you provide the pseudo-code of the i-LN you proposed? This would be helpful for us to quickly get started with the implementation and accurately reproduce the results.

---

> ### Author Response · Authors · 2025-11-22
>
> We sincerely thank the reviewer for the careful reading and helpful comments.
>
> We appreciate for the positive evaluation of our motivation, theoretical grounding, and plug-and-play design. We’re especially glad the reviewer found our analysis of low-level-vision–specific normalization issues to be clear and convincing, and appreciated the simplicity of i-LN as a targeted, easily verifiable fix rather than a direct transplant from other tasks.
>
> Below is a pseudo code of our implementation. Here, we simply consider the Attention Block, but extending it to FFN or others is straightforward.
>
> ```
> def AttnBlock_vanillaLN(x):
>   # x.shape == (N, L, C)
>
>    skip = x
>
>    mu, sigma = norm_stats(x, dim=C)        # Vanilla LN: Per-token over channels
>    x = (x - mu) / (sigma + eps)            # Normalize: mu.shape == N,L,1 | sigma.shape == N,L,1
>    x = x * gamma + beta                    # Affine: gamma.shape == 1,1,C | beta.shape == 1,1,C
>
>    x = Attn(x)
>
>    return x + skip
> ```
> ```
> def AttnBlock_iLN(x):
>   # x.shape == (N, L, C)
>
>    skip = x
>
>    mu, sigma = norm_stats(x, dim=(L, C))   # Spatially holistic: over all-tokens+channels
>    x = (x - mu) / (sigma + eps)            # Normalize: mu.shape == N,1,1 | sigma.shape == N,1,1
>    x = x * gamma + beta                    # Affine: gamma.shape == 1,1,C | beta.shape == 1,1,C
>
>    x = Attn(x)
>    x = x * sigma                           # input-adaptive rescaling
>
>    return x + skip
> ```
> - In cases where LN is not coupled with other operations (e.g., the last LayerNorm), we can simply substitute it with an Affine Transformation Layer without any normalization operation (i.e., only ```gamma```, ```beta```).
> - Image are normalized with DIV2K stats before forwarding to the network with i-LN.
> - Re-introducing the mean (e.g., ```x = x * sigma + mu```) that is subtracted during the normalization step might also be a straightforward extension of our i-LN; but simply has no empirical effects since feature from our i-LN is already near zero (Fig.13.b)
>
>
> ---
>
>
> Additionally, please refer to Appendix.E-F where we have included insightful experimental results thanks to the constructive reviews. We sincerely appreciate the time and effort for the reviews. (**Updated regions in the manuscript are marked in blue.**)
>
> - Additional visual results:
>     - (1) visual results for benchmarks (Fig.16-23)
>     - (2) visual comparison between LN, i-LN, LN*, LN+Rescaling (RPE in Fig.12, SR result it Fig.13)
>     - (3) visual comparison for real-world SR (Fig.15)
> - Additional experiments regarding robustness of i-LN:
>     - (1) analysis over training configuration variation (updated Appendix.E.1, Fig.14)
>     - (2) analysis over multiple random seeds  (updated Sec.3.5.3, Fig.11)
>     - (3) scores for real-world SR (Appendix.E.2, Tab.9, Fig.15)
>
> We hope these additional analyses are useful in further clarifying our contribution and in supporting the reviewer’s final assessment of the our work.

---

### Official Review · Reviewer_SRbU · 2025-10-28

**Soundness:** 3
**Presentation:** 4
**Contribution:** 3
**Rating:** 6
**Confidence:** 3

**Summary:**

This paper investigates why Image Restoration (IR) Transformers often show unstable training behavior when using standard Layer Normalization (LN). The authors find that LN causes feature magnitudes to diverge and channel entropy to collapse, indicating a fundamental mismatch between LN and dense prediction tasks.
To address this, they propose i-LN, a simple drop-in replacement that applies spatially holistic normalization and input-adaptive rescaling. i-LN stabilizes training and improves performance across multiple IR tasks (super-resolution, denoising, deblurring, and artifact removal) and Transformer architectures.

**Strengths:**

1. Novel insight into a fundamental issue – The paper is the first to thoroughly diagnose the instability and abnormal training dynamics of IR Transformers caused by LN. The observations of feature divergence and entropy collapse are well-supported and enlightening.

2. Strong theoretical grounding – The authors formalize the notion of structure-preserving transformations and mathematically show that LN fails to maintain spatial isomorphism, whereas the proposed i-LN satisfies these properties.

3. Simple yet impactful method – i-LN is conceptually straightforward, easily implemented, and compatible with existing architectures without any retraining overhead.

**Weaknesses:**

1. The theoretical analysis is insightful but somewhat idealized — it assumes a simplified setting without the learnable affine parameters (β, γ) and does not account for the nonlinear activations that follow normalization. While this abstraction helps clarify the core argument, it may limit the generality of the conclusions in practical Transformer architectures.
2. Qualitative evaluation could be expanded – The visual examples are convincing, but adding perceptual metrics (e.g., LPIPS, FID) or user studies would make the improvements more tangible.

**Questions:**

see the weaknesses.

---

> ### Author Response · Authors · 2025-11-22
>
> We thank the reviewer for their time and effort in evaluating our work. We especially appreciate the careful reading and the recognition of both our diagnostic contribution and the simplicity and impact of i-LN. We are glad that the core observations (feature magnitude divergence and channel-entropy collapse) and the structure-preservation framing came through clearly.
>
> ---
>
> ### Concern Regarding Abstracted Proof
> We appreciate the reviewer’s concern regarding the simplified setting. Our theoretical analysis indeed omits the affine parameters ($\gamma$, $\beta$). This is in order to isolate the effect of the normalization operation itself.
>
> Since $\gamma/\beta$ are learnable parameters applied after normalization and shared across all tokens, we do not view them as the primary source of information loss or the resulting severe instability. Rather, they can be grouped with other learnable modules following normalization (e.g., attention/FFN stacks with nonlinear activations), whose role is to adaptively transform features through training.
>
> Such learnable post-normalization components can naturally realize information-preserving transformations when beneficial; importantly, this does not require the disastrous feature-magnitude divergence observed under vanilla LN. For example, even for nonlinearities like ReLU, a simple global shift provides a trivial bypass, indicating that information can be preserved without inducing extreme feature behavior.
>
> In fact, our empirical evidence suggests an even stronger point: as shown in Sec. 3.4.2 and Fig. 8, $\gamma/\beta$ can be used to counteract LN-induced information loss. This aligns with our intention behind the simplified proofs and reinforces our main claim that the mechanism of vanilla LN are the core driver of information loss and the corresponding instability.
>
>
> ---
> ### Requires More Qualitative Comparison
>
> We agree with the reviewer that the qualitative analysis could be expanded. Accordingly, we have included additional visual comparison in the Appendix as specified below. We thank the reviewer for the helpful suggestion.
>
> - (1) visual results for benchmarks (Fig.16-23)
> - (2) visual comparison between LN, i-LN, LN*, LN+Rescaling (RPE in Fig.12, SR result it Fig.13)
> - (3) visual comparison for real-world SR (Fig.15)
>
> ---
>
> Additionally, we would like the reviewer to please refer to Appendix.D-F where we have included further analysis thanks to the constructive reviews. (**Updated regions in the manuscript are marked in blue.**)
>
> - Additional experiments regarding robustness:
>     - (1) analysis over training configuration variation (updated Appendix.E.1, Fig.14)
>     - (2) analysis over multiple randomseeds (updated Sec.3.5.3, Fig.11)
>     - (3) scores for real-world SR (Appendix.E.2, Tab.9, Fig.15)
>
> We hope these additions are useful in addressing further potential concerns and in supporting the reviewer’s final assessment of the paper. At last, we would like to thank the reviewer again for their time and effort in reviewing our work.

---

> > ### Comment · Reviewer_SRbU · 2025-11-28
> > **Post-Rebuttal Assessment**
> >
> > I thank the authors for their detailed rebuttal and the clarifications presented. The additional explanations resolve my principal concerns and improve the overall clarity of the submission. Taking the rebuttal into account, I will keep my initial assessment unchanged.

---

> > > ### Author Response · Authors · 2025-12-02
> > >
> > > Thank you for your thoughtful follow-up and for taking the time to review our rebuttal.
> > > We appreciate your careful consideration of our clarifications and are glad to hear that they addressed your main concerns.
> > > We respect your decision to keep the initial assessment unchanged, and we are grateful for your constructive feedback, which helped improve the clarity of the submission.
> > >
> > > Due to circumstances, we are unfortunately unable to continue further discussion at this stage, but we sincerely appreciate the time and attention you have dedicated to our work.

---

### Official Review · Reviewer_EZi7 · 2025-10-31

**Soundness:** 3
**Presentation:** 3
**Contribution:** 2
**Rating:** 4
**Confidence:** 5

**Summary:**

This paper studies the effect of Layer Normalization (LN) in Transformer-based image restoration networks, focusing on the observed instability and feature magnitude divergence during training. The authors analyze how the per-token LN operation may disturb spatial correlations and feature statistics, and propose a variant called i-LN, which normalizes across both spatial and channel dimensions with an input-dependent scaling term. Experiments across several image restoration tasks (SR, DN, DR, CAR) show modest but consistent improvements in stability and quantitative performance. Overall, the work offers a careful empirical examination of normalization behavior in low-level Transformer models, though the methodological change is relatively minor.

**Strengths:**

1. The paper focuses on a clear and well-defined problem — the instability of Layer Normalization in image restoration Transformers — and provides intuitive analysis to support it.

2. The proposed i-LN module is conceptually simple and easy to implement, making it practical for existing Transformer-based IR frameworks.

3. Experiments across several restoration tasks show consistent but moderate improvements, indicating the method’s general usefulness.

4. The paper is clearly written and well-organized, making the motivation and results easy to follow.

**Weaknesses:**

### Main Concerns:
1. Since the proposed i-LN is designed to stabilize feature statistics across varying inputs, it would be informative to evaluate its effectiveness in an All-in-One restoration setting, where a single model must generalize across multiple degradation types. This could further validate the claimed robustness and statistical consistency advantages.

2. While the proposed i-LN improves feature stability by normalizing across spatial and channel dimensions, this inevitably increases the computational scope. The paper claims negligible overhead but provides no quantitative analysis to support this; it would be helpful to report the actual runtime or FLOPs to assess the trade-off between stability and efficiency.

3. For a low-level vision paper, the visual comparisons are too limited and small in scale. Given that image restoration quality is best evaluated through perceptual inspection, more visual comparison samples are expected to be included in at least the appendix or supplementary materials, but this is missing.

4. While the proposed i-LN is claimed to improve feature stability and robustness, the paper does not include any evaluation on real-world degradations, which are common and practically relevant in image restoration. Since the method is designed to better preserve low-level statistics, testing it on real-world datasets would be important to demonstrate its practical effectiveness and generalization beyond synthetic benchmarks.

5. Despite the solid analysis, the performance gain is modest, which may limit perceived impact.

6. The method is incremental in implementation—essentially a normalization variant—though well justified.

7. Computational overhead or efficiency impact of holistic normalization is not deeply discussed.

8. Could include more discussion on generalization beyond image restoration tasks (e.g., classification, segmentation)?.

### Minor Concerns:
1. The computation resources are expected, for example, the authors only mentioned in Line 236 via A6000 GPUs, but how many for each experiment?

2. The code of the proposed method is expected to be released.

3. The term “IR Transformer” is not clearly defined — since the architecture is largely identical to standard vision Transformers, the authors should clarify whether the analysis truly targets image restoration–specific behavior or applies more broadly to general Transformer models.

**Questions:**

1. Could the authors provide quantitative evidence of efficiency, such as runtime, FLOPs, or memory cost, to support the claim that the proposed holistic normalization incurs negligible computational overhead?

2. Since i-LN aims to stabilize feature statistics, have the authors considered evaluating it under an All-in-One or real-world degradation setting to demonstrate robustness across diverse degradation types?

3. The paper focuses on Transformer-based IR models — do the authors expect i-LN to generalize to non-Transformer or hybrid CNN–Transformer architectures? If so, could preliminary results or insights be shared?

4. The analysis emphasizes improved stability and entropy preservation, but could the authors quantify this improvement (e.g., through variance reduction, entropy scores, or training convergence curves) to better support these claims?

5. Given that the proposed method is relatively simple, could the authors discuss whether i-LN interacts with other normalization techniques (e.g., InstanceNorm, GroupNorm) or whether similar stability effects could be achieved by alternative normalization formulations?

Please also refer to the **Weaknesses**.

---

> ### Author Response · Authors · 2025-11-22
>
> We sincerely appreciate the reviewer’s thorough and constructive feedback, which helped us strengthen both the analysis and empirical validation of the paper.
>
> ---
>
> ## 1. Stability and Robustness of $\textit{i}$-LN
>
> ### 1.1. Realworld Super-Resolution (updated Appendix.E.2, Tab.9, Fig.15)
> We follow the RealESRGAN pipeline and synthesize the DF2K dataset and synthesize standard benchmarks (Set5, Set14, B100, Urban100, Manga109) for testing. We report scores across five random seeds.
>
> | $\textcolor{grey}{\text{PSNR (Avg. over 5 Runs)}}$       | Set5 |  Set14 |  B100  |  Urban100  |  Manga109  |
> | ------------ | ----- | ------------ | ----- | ------------ | ----- |
> | RealHAT + LN (baseline)           | 25.68 | 24.53 | 24.35 | 21.95 | 23.53 |
> | RealHAT + $\textit{i}$-LN (Ours)         | 25.83 | 24.71 | 24.43 | 22.14 | 23.80 |
>
> $\textcolor{gray}{\text{-- Table: Quantitative results for real-world SR.}}$
>
> As shown above, our $\textit{i}$-LN significantly outperforms vanilla LN on all benchmarks, further confirming the robustness and practical applicability of $\textit{i}$-LN. Please refer to Tab.9 for PSNR/SSIM scores for each run, and Fig.15 for visual examples.
>
> These results highlight that our $\textit{i}$-LN is not only theoretically grounded but also practically beneficial in challenging real-world restoration scenarios.
>
> ---
>
> ### 1.2. Training curves over various training configuration (updated Appendix.E.1, Fig.14)
> We train the HAT model under a wide range of batch sizes, from 2 to 8, to assess training stability. As shown in Fig.14, our $\textit{i}$-LN consistently achieves higher PSNR/SSIM than the LN baseline in every batch size setting; and the performance gap is also stable across all batch size.
>
> These results demonstrate that the benefit of our $\textit{i}$-LN is not tied to a particular training setup, and that its stability advantages persist even under extremely training configurations.
>
> ---
>
> ### 1.3. Training curves over multiple runs with different random seeds (updated Sec.3.5.3, Fig.11)
> We train the HAT model under multiple runs with various random seeds. As shown in Fig.11, the **variance** across seeds is significantly lower for $\textit{i}$-LN (in terms of both training dynamics and PSNR scores) even in this hard training setting, while **vanilla LN often suffers significant fluctuation**. This indicates that $\textit{i}$-LN leads to a more reliable and stable training. Also, our $\textit{i}$-LN consistently achieves higher PSNR/SSIM than the vanilla LN baseline in every run.
>
> These results demonstrate that our $\textit{i}$-LN provides a more reliable optimization landscape, reducing susceptibility to randomness in initialization or data ordering, which is an important practical advantage for training IR networks.

---

> ### Author Response · Authors · 2025-11-22
>
> ## 2. Lack of Computational Cost Analysis
>
> In short, the computational cost overhead is neglectable by introducing i-LN. Specifically, in terms of Multi-Adds, **HAT(+LN) requires 103.7G** and changing it to **HAT(+i-LN) only needs an additional 0.063G (or less) overhead**.
>
> Below, we would like to provide the analysis of the computational cost of i-LN in three folds: (1) the normalization step (w/o affine), (2) the affine transformation step, and (3) the input-adaptive rescaling step. In short, i-LN only induces neglectable computational cost since it only requires a single additional scalar multiplication per each normalization step.
>
> - **Normalization (Identical Cost).**
> We sincerely appreciate the constructive review. However, we would like to politely disagree with the reviewer's claim of "increased computational scope" regarding the spatially holistic normalization scheme. Although i-LN normalizes across both spatial and channel dimensions, this does not increase the computational scope. Both LN and i-LN must read the exact same number of feature values (H×W×C), and both require a single pass over all elements. LN normalizes C elements but repeats this H×W times, while i-LN normalizes HWC elements once. Those are exactly the same total O(HWC) compute.
>
> - **Affine Transformation (Identical Cost).**
> The affine transformation for both LN and i-LN is simply a per-element multiplication and an addition, thus both have identical computational cost.
>
> - **Rescaling (i-LN requires more cost, but is neglectable).**
> In essence, compared to LN, the proposed i-LN only requires a single additional scalar multiplication introduced by the input-adaptive rescaling step, coupled with each normalization operation. Given the heavy computation in IR Transformers, particularly attention and FFN operations, this extra single scalar multiplication is negligible in practice.
>
> - **(Exact Multi-Adds)** More concretely, considering a 64x64 input under the default HAT configuration (embedding dimension 180 with 86 normalization layers), the baseline model with LN requires 103.7G Multi-Adds (referring to the official HAT paper). i-LN introduces only one extra scalar multiplication coupled with each normalization layer, which corresponds to: 64 x 64 x 180 = 737,280 Multi-Adds per rescaling operation. With 86 normalization layers, the total additional cost becomes: 86 x 737,280 = 0.063 G Multi-Adds, clearly negligible relative to the entire computations (103.7 G Multi-Adds).
>
> Further in practice, this can be accelerated by modulating the projection layer weight instead of the feature itself. Also, we can remove the entire normalization layer itself in cases where the norm-and-rescale does not encapsulate any operation (e.g., positions as the final LayerNorm or patchify), further reducing the tiny gap. Overall, i-LN does not induce any meaningful computational cost compared to LN, while substantially stabilizing feature statistics and improving reconstruction performance.

---

> ### Author Response · Authors · 2025-11-22
>
> ## 3. Relationship with other Normalization
>
> We thank the reviewer for sharing the constructive insights into the necessity of comparison against other normalization or regularization techniques. In **Sec.3.1, Tab.1, Fig.4, Tab.8** of the original manuscript, we have compared the following methods: 1) mainly LN and i-LN, and 2) other normalization schemes as BatchNorm, InstanceNorm, RMSNorm and 3) recent techniques to stabilize training as LayerScale, ReZero and 4) regularization as GradClip, KL Divergence, and 5) also the configuration without any normalization (None); leading to analysis and comparison across a total of 10 normalization relevant schemes.
>
>
> - **(Stability Effects)** As can be seen in Fig.4, any type of per-token normalization scheme (even with improved techniques, such as LayerScale) suffers from diverging feature scales, while any type of non-per-token (i.e., spatially holistic) normalization scheme shows stable feature scales (including the configuration without any norm).
>
> - **(Performance)** As can be seen in Tab.1, normalization schemes that act in a per-token manner always perform worse than the spatially holistic variants. The exceptional case is the None configuration (no normalization across the entire network), which simply fails to converge.
>
>
>
> | Method       | PSNR|
> | ------------ | ----- |
> | HAT+LN (Tab.1)           | 26.55 |
> | HAT+InstanceNorm (Tab.1) | 27.02 |
> | HAT+GroupNorm    | 27.08 |
> | HAT+i-LN (Tab.1)         | 27.17 |
> **$\textcolor{gray}{\text{-- Table: Additional experiment on GroupNorm, also leading to significant performance gain, aligning with our analysis.}}$**
>
> - **(Additional experiment: GroupNorm)** We appreciate the helpful suggestion. Accordingly, we have additionally performed an analysis with GroupNorm (now total analysis across 11 configurations), which falls within the spatially holistic variants we have analyzed in Tab.1, Fig.4 and Sec.3.1. The identical behavior can be observed: GroupNorm shows notably stable feature scales (\~2.0x10^1) compared to LN (\~5.6x10^6), while performing significantly better than LN (but still worse that our i-LN).
>
> ---
>
> ## 4. Additional Visual Examples
> We thank the reviewer for the helpful suggestion.
> We have included additional visual results for (1) benchmarks in Fig.16-23 and (2) for comparison between LN, i-LN, LN*, LN+Rescaling in Fig.13, and (3) for qualitative evaluation under real-world settings in Fig.15.
>
> ---
>
> ## 5. Limited novelty of i-LN and marginal performance gain
> We sincerely appreciate the constructive review. However, we would like to kindly note that our primary contribution lies in the in-depth analysis of the training dynamics of IR Transformers, rather than proposing a technically complicated method for peak performance.
>
> Through our analysis, we identify and characterize the problematic behaviors induced by conventional LayerNorm, which remains the default normalization choice in Transformer-based IR architectures. Our findings reveal fundamental misalignments between per-token normalization and the statistical requirements of IR tasks, shedding new light on previously overlooked dynamics.
>
> The intentional simple design of i-LN is to validate the effectiveness of addressing the issues exposed through our analysis, rather than aiming for peak performance by introducing a complex or heavily engineered module. This simplicity allows us to isolate and directly test these two key factors, ensuring that the improvements arise from the principles identified in our investigation rather than architectural overfitting.
>
> Despite this simple modification, our method leads to significant performance gain, especially for SR and Deraining (e.g., PSNR +2.42 dB on SwinIR-Test100) tasks, and also meaningful performance gain for Denoising and Compression Artifact Removal tasks. Regarding the concerns of the reviewer of small performance gain in several scenarios, further tailoring i-LN for each restoration task for peak performance can indeed be a valuable direction for future work. We sincerely appreciate the constructive review.

---

> ### Author Response · Authors · 2025-11-24
>
> ## 6. Applicability of i-LN in CNN-Transformer Hybrid Architectures
>
> We would like to thank the reviewer for the constructive suggestion. Accordingly, we have conducted an additional experiment regarding applicability of i-LN in the CNN-Transformer Hybrid architecture.
>
> To simulate a CNN–Transformer hybrid architecture, we augment each FFN block in $\text{HAT}_1$ with a 3×3 depth-wise convolution (DWConv) layer, effectively replacing the linear-only FFN with a DWConv-FFN structure. Accordingly, the canonical sequence $Norm - Attention - Norm - MLP $ becomes $Norm - Attention - Norm - DWConvFFN$; mimicing the conventional Conv-Transformer hybrid architecture.
>
>
> | PSNR($\textcolor{grey}{+Gain}$)                   | Set14 | BSD100 | Urban100 |
> |:-------------------------|:------|:-------|:---------|
> | $\mathrm{HAT}_{1}$ +i-LN (Ours) | 29.01 | 27.84  | 27.17    |
> | +DWConvFFN                   | 29.28 $\textcolor{grey}{(+0.26)}$ | 27.85 $\textcolor{grey}{(+0.01)}$  | 27.22  $\textcolor{grey}{(+0.05)}$  |
>
> | PSNR($\textcolor{grey}{+Gain}$) | Set14 | BSD100 | Urban100 |
> |:-------------------------|:------|:-------|:---------|
> | $\mathrm{HAT}_{1}$ +LN  (Baseline) | 28.79 | 27.68  | 26.55    |
> | +DWConvFFN                   | 29.18 $\textcolor{grey}{(+0.39)}$ | 27.79 $\textcolor{grey}{(+0.11)}$ | 26.88  $\textcolor{grey}{(+0.33)}$  |
>
> **$\textcolor{gray}{\text{-- Table: Analysis on other architectures.}}$**
>
> ---
>
> - **(More Performance Gain for vanilla LN)** Importantly, **performance gain are more significant in the vanilla LN variant, which supports our claim that networks with vanilla LN often suffer in modeling spatial relationship**; thus, introducing Conv layers helps more. In contrast, our i-LN variant shows smaller performance gain since our network is already successfully capturing the local spatial relationship.
>
> - **(Same Feature Divergence Trend)** Even in this hybrid architecture, we have observed the same feature divergence phenomenon. The Conv augmented HAT(+LN) exhibits feature magnitudes diverging up to $>1.5\times10^7$, whereas our Conv augmented HAT(+i-LN) variant maintains consistently stable statistics throughout training (converging to near 1), aligning with our core intuition: LayerNorm is the key problem.
>
> - **(Final Performance)** Despite the vanilla LN variant benefiting more from Conv layers, our i-LN variant still outperforms in all benchmarks.
>
> Overall, this experiment shows that 1) i-LN is superior even in the Conv-Transformer hybrid architecture, and 2) vanilla LN indeed suffers from modeling (local) spatial relationship (regarding the performance gain w/ Conv layers). Thus, we conclude that benefits of i-LN extend beyond pure transformer blocks.
>
> ---
>
> ## Minor
> - Regarding the number of GPUs, we use 1 GPU for the smallest variant HAT$_1$; and 4 GPUs for the intermediate variant HAT$_2$; and 8 GPUs for the largest variant HAT$^\dagger$. In all cases we have used NVIDIA A6000.
> - We are planning to released the code after official publication.
> - Our $\textcolor{red}{i\text{-LN}}$ ($\textcolor{red}{\text{i}}$mage restoration tailored $\textcolor{red}{\text{LN}}$) primarily focuses in  Vision Transformers coupled with Image Restoration tasks. IR Transformers indeed have similar architectures to plain ViTs. However, each token directly indicates individual pixels (i.e., no patchify) and requires to strictly preserve the 1) low-level feature statistics and 2) per-pixel inter-token relationship of the given input image; a IR task specific property, which we have mainly tackled.

---

### Official Review · Reviewer_Msrh · 2025-11-01

**Soundness:** 3
**Presentation:** 3
**Contribution:** 2
**Rating:** 4
**Confidence:** 2

**Summary:**

The paper address the image reconstruction task in vision domain.
The authors found that image reconstruction using transformers with vanilla layer norm lead to feature magnitudes divergence and  channel entropy collapses.
The authors further analyze the patterns by relaxing the constraints of layer norm, i.e., per-token normalization and input-independent scaling. Specifically, vanilla layer norm breaks spatial correlation due to the per-token normalization and discard input-specific statistics due to the input-independent scaling, which are critical for image restoration tasks. To handle these issues, the authors propose i-LN, which normalizes holistically across space and channels and re-scales each block by the input’s own standard deviation to recover global scale. Experimental results qualitatively and quantitatively show that i-LN obtain better PSNR and SSIM on super-resolution, deraining, denoising, and JPEG artifact removal than vanilla LN on various benchmarks.

**Strengths:**

* Easy to read
* Simple but intuitive claims
* Extensive experiments

**Weaknesses:**

* Major concern
    * While the authors demonstrated the strengths of the proposed i-LN over vanilla layer norm (LN) and spatially holistic layer norm (LN*), the experiments only show the comparisons with vanilla LN and other normalization approaches. If the provided claims are valid, I guess the superiority among LN, LN*, and i-LN on the image restoration problem should be LN < LN* < i-LN. Could you further validate this superiority, not only over specific benchmarks but over various benchmarks?
    * While the authors provided channel entropy collapse comparison among LN, LN*, and i-LN in Fig. 8, it seems not sufficient since the volume for this comparative analysis is too light than the overall volume of the experiments. And still there's no qualitative or quantitative performance comparison among LN, LN* and i-LN, while both are necessary.
    * Suggestion
        * I recommend to supplement these experimental results. Since the results for LN and i-LN are already secured, only results for LN* are necessary for the following comparisons:
            * relative position embedding comparison (Fig. 9) among LN, LN* and i-LN
            * Qualitative results comparisons (Fig. 6) among LN, LN* and i-LN
            * Quantitative comparisons (Table 1, 2) for at least two datasets (e.g., Set14 and BSD100) among LN, LN* and i-LN
    * I'll raise my rating if this concern is resolved
* Explanation for the derivation of feature divergence in Figure 4 and other experimental results in some place (e.g., the paragraph in lines 260-267) would be helpful

**Questions:**

* Why were three type of model implementations necessary?
* I recommend to reorganize the location of Table 5 in appendix. There would be better position than the top of the first page of Appendix

---

> ### Author Response · Authors · 2025-11-22
>
> We sincerely appreciate the reviewers’ time and constructive feedback.
> Below, we clarify the concerns raised and provide additional analyses and explanations regarding comparison between LN, LN*, i-LN, and also LN+Rescaling.
>
> ---
>
> ## 1. Comparison between LN, LN*, i-LN (Sec.3.3, Appendix D)
>
> As the reviewer pointed out, although our paper primarily highlights the superiority of i-LN over LN, a more thorough analysis of the relationship LN < LN* < i-LN is indeed necessary. We address this in three folds.
>
> ---
>
> ### 1.1 Quantitative Scores (Tab.3)
> Tab.3 in our original manuscript provides the quantitative scores that compare LN, LN*, i-LN, and additionally LN+Rescaling (i.e., i-LN without spatial holisticness). Each is as follows: LN is Tab.3.Idx1, LN* is Tab.3.Idx3, i-LN is Tab.3.Idx4, and additionally LN+Rescale is Tab.3.Idx.2.
>
> As the reviewer has expected, the scores follow the order of LN < (also LN+rescale) < LN* < i-LN. This verifies the necessity of both 1) spatial holisticness based on acquiring normalization factors through the entire spatio-channel dimension and 2) input dependent statistic preservation by rescaling.
>
> We acknowledge that our current presentation may be confusing. In the camera-ready version, we will revise the text to more clearly describe the ablation settings and naming conventions (LN, LN*, i-LN), ensuring the relationships and purposes of each variant are unambiguous. We appreciate the reviewer’s constructive suggestions.
>
> ---
>
>
> ### 1.2. Relative Position Embedding (RPE) Comparison (Updated Fig.12)
> In Fig.12, we added an extended visual comparison of RPEs among LN, LN*, i-LN and also LN+Rescaling.
>
> - The Vanilla LN variant exhibits highly noisy RPE structures throughout the entire training process, especially in the RPEs from the deep layers.
>
> - The LN* variant occasionally reveals faint global structures in the deep layers, but these patterns remain weak and are overshadowed by considerable noise.
>
> - The LN+Rescaling variant shows improved RPE structures in deeper layers, yet its shallow-layer embeddings remain unstable and noisy, even after the full training (500K).
>
> - Our i-LN consistently produces substantially clearer and more structured RPE maps, with significantly reduced noise across both shallow and deep layers, and across all training stages. The strong spatial coherence visible in the embeddings aligns with the quantitative improvements reported in Tab.3, where i-LN achieves the highest performance among all evaluated variants.
>
> ---
>
>
> ### 1.3. Qualitative Comparison for the SR result (Updated Fig.13)
> In Fig.13, we additionally provide visual comparisons of the SR outputs, with networks each employing LN, LN*, LN+Rescale and i-LN. As can be seen, i-LN yields the sharpest and cleanest recovery of fine-grained details. Vanilla LN often results in blurry structures and fails to reconstruct high-frequency components. The ablated variants (LN* and LN+Rescaling) show noticeable improvements over LN but still exhibit mild degradations such as oversmoothing or loss of local textures.
>
> Overall, the qualitative comparisons provide visual evidence that both components of i-LN (spatial holisticness and input-adaptive rescaling) are essential.
>
>
> ----
>
> ## 2. Derivation of Feature Divergence
> Following EDMv2 [1], we report the squared mean of the full feature tensor (NCHW), which captures both mean and variance in a single statistic (Line 83). For example, a feature following a standard normal distribution is expected to have a squared mean near 1; and any drift (i.e., mean is not zero) or increased variance are reflected as high square mean values. We updated visualization of linear-scale mean and variance in Fig.11.
>
> [1] Karras, Tero, et al. "Analyzing and improving the training dynamics of diffusion models." Proceedings of the IEEE/CVF Conference on Computer Vision and Pattern Recognition. 2024.

---

> > ### Author Response · Authors · 2025-11-22
> >
> > ## 3. Three Implementation Settings
> > Our paper aims to provide extensive experimental evidence to reveal the training dynamics of IR Transformers. Thus, we include experiments across diverse network configurations, degradation types, degradation strengths, and more.
> >
> > Due to computational limitations, the experiments in Tab.1–2 (the $\text{HAT}_1$ setting) use a reduced training budget. To rigorously evaluate the effectiveness of each component, we expand the computational budget in a more demanding configuration (the $\text{HAT}_2$ setting). This allows us to confirm that the improvements of our method persist under stronger training conditions and are not due to the limited-resource regime used in earlier experiments. Finally, we verify that our method also generalizes to the computationally intensive classical SR benchmark setting by comparing with the officially released checkpoints of prior works (the $\text{HAT}^\dagger$ setting). Collectively, these three settings seek for a balance between computational practicality and thorough evaluation.

---

### Author Response · Authors · 2025-11-26
**Official Comment by Authors (Summary of Discussion)**

Dear PCs, SACs, ACs, and Reviewers.

We sincerely appreciate the time and effort for reviewing our paper.
- Below, we provide a brief summary of 1) our key contribution and 2) the discussion.
- For convenience during the discussion period, most updates are placed in the Appendix (marked as $\textcolor{blue}{blue}$). These will be integrated into the main manuscript for the camera-ready version.

Best Regards,

$i$-LN Authors.

---
---

### Key Contribution

- First systematic analysis of the training dynamics of Image Restoration Transformers, revealing key failure modes of existing normalization practices.
- Theoretical analysis showing how structure preservation and LayerNorm behavior impact IR models, followed with extensive experiments.
- A simple plug-and-play normalization design that yields consistent and significant improvements across IR tasks, including real-world restoration settings.
---

### Reviewer 1 (Msrh, Rating4)
> **Reviewer Msrh has stated to $\textcolor{red}{\text{raise the rating}}$ if concern-1 is resolved.**
>
> *''I'll raise my rating if this concern is resolved.''*

- **Concern 1**: Needs comparison between LN (baseline), i-LN (ours), and LN*. Specifically, provide 1) quantitative scores, 2) qualitative results, 3) RPE visualization.
- **Response 1**: This corresponds to the ablation study.
    - (1-1) Quantitative scores (PSNR) are already in Tab.3 of the original submitted manuscript.
    - (1-2) Quantitative analysis in terms of Entropy is already in Fig.7 of the original submitted manuscript.
    - (2) We updated qualitative results in Fig.13 during the discussion period.
    - (3) We updated RPE visualization in Fig.12 during the discussion period.


No other major concerns are stated.

---

### Reviewer 2 ( EZi7, Rating4)
> **We did not receive any post-rebuttal feedback yet.**
> - **However, concerns are addressed as below.**
> - **Reviewer EZi7 appreciates 1) the well-defined problem; 2) intuitive analysis to support it; and 3) the strong applicability of our $i$-LN.**

- **Concern 1**: To validate stability/robustness of $i$-LN, provide 1) variance and training curves or 2) real-world examples.
    - **Response 1**:
        - (1) Added training curves / variance (updated Sec.3.5.3, Fig.11, Appendix.E.1, Fig.14)
        - (2) Added real-world example (updated Appendix.E.2, Tab.9, Fig.15)

- **Concern 2**: Lack of Computational Cost Analysis
    - **Response 2**: The cost is negligible. Changing from LN to $i$-LN only requires a single scalar multiplication.
- **Concern 3**: Missing analysis about the relationship with other Normalization
    - **Response 3**: Already provided in Sec.3.1 (Tab.1, Fig.4-5). This is a central part of our analysis.
- **Concern 4**: Limited Visual Examples
    - **Response 4**: Updated visual examples:
        - (1) benchmarks (Fig.16-23)
        - (2) comparison between LN, i-LN, LN*, LN+Rescaling (Fig.12, Fig.13)
        - (3) real-world SR (Fig.15)
- **Concern 5**: Limited Novelty and Marginal Performance Gain
    - **Response 5**: Our contribution lies in the in-depth analysis. The simple design of i-LN is in order to isolated experimental factors and validate our claims; not aiming for peak performance. Despite this, performance gain is also significant  (e.g., PSNR +2.42 dB on SwinIR-Test100)
- **Concern 6**: Applicability of $i$-LN to other Architecture
    - **Response 6**: Added experiment with Conv-Attention Hybrid architecture. (Please see the table in the Author Comment. Search for ```DWConvFFN```).

---

### Reviewer 3 (SRbU, Rating 6)
> **Reviewer SRbU has stated to $\textcolor{red}{\text{keep the original positive score}}$ (rating 6):**
>
> **The reviewer also appreciates the novel insights and the strong theoretical analysis.**
>
> *''The additional explanations resolve my principal concerns ... I will keep my initial assessment unchanged.''*

- **Concern 1**: Abstracted Proof.
    - **Response 1**: The Affine transformation is indeed abstracted. This is in order to primarily focus on the normalization operation itself, and also since it is unlikely that learnable affine transformations disrupt spatial correlations.

- **Concern 2:** Limited qualitative evaluation.
    - **Response 2:** Updated visual examples:
        - (1) visual results for benchmarks (Fig.16-23)
        - (2) visual comparison between LN, i-LN, LN*, LN+Rescaling (RPE in Fig.12, SR result in Fig.13)
        - (3) visual comparison for real-world SR (Fig.15)



---

### Reviewer 4 U1uB (Rating 6)
> **Reviewer U1uB has stated that $\textcolor{red}{\text{no weaknesses are identified}}$ with an initial $\textcolor{red}{\text{positive rating 6}}$.**
>
> **The reviewer also appreciates (1) the simplicity and the (2) potential impact of our $i$-LN in the field of low-level vision:**
>
> *''No obvious weaknesses are identified ...  may become the standard approach for designing network structures in the field of low-level vision''*

- No concern.

---

### Meta-Review · Area_Chair_4WLm · 2026-01-10

**Summary:**

This paper investigates the effect of Layer Normalization (LN) in Transformer-based image restoration networks, focusing on the observed instability and feature magnitude divergence during training. Two reviewers are slightly negative, while the other two are slightly positive; nevertheless, all four reviewers praised the novel insight into this fundamental problem. The authors provided detailed and persuasive responses, and it is very likely that the two reviewers on the negative side would be satisfied with the technical clarifications and additional evidence. Considering the overall evaluations and the rebuttal, AC believes this paper is above the acceptance bar and recommends acceptance as a poster.

**Reviewer Concerns:**

Reviewer Msrh (Marginally below acceptance) mainly questioned whether the ablation evidence really supports the claimed ordering LN < LN* < i-LN, noting that prior results lacked direct quantitative/qualitative comparisons among these three, and also asked for clearer explanation of the derivation behind the feature divergence analysis. In rebuttal, the authors largely resolved the core request by pointing to Tab. 3 for LN/LN*/i-LN comparisons, adding/expanding qualitative SR results and RPE visualizations, and clarifying how their divergence statistic is computed. So, the technical concerns have mainly been resolved.

Reviewer EZi7 (Marginally below acceptance) worried about robustness/stability validation, missing concrete efficiency/cost numbers, and insufficient visual/real-world evidence, while also flagging that the method is incremental and the gains can look modest. The rebuttal substantially addressed most points by adding training curves/variance across seeds and batch sizes, including a real-world SR experiment, expanding visual results, and providing a quantitative Multi-Adds overhead estimate arguing negligible cost.

Reviewer SRbU (Marginally above acceptance) was generally positive on the insight and theory, but raised that the theoretical analysis is somewhat idealized, and suggested expanding qualitative evaluation (and optionally perceptual metrics). The authors’ rebuttal justified the abstraction as isolating normalization effects, argued learnable affine/post-norm components are not the driver of the observed instability, and added substantially more qualitative/robustness material. The reviewer stated post-rebuttal that score will keep unchanged.

Reviewer U1uB (Marginally above acceptance) reported no major weaknesses, emphasizing the simplicity and potential impact of i-LN.

**Reviewer Scores:**

Reviewer Msrh explicitly mentioned that, given satisfactory response, the score will be raised. Considering that the rebuttal sufficiently resolved this reviewer's concerns regarding ablation evidence really supports the claimed ordering LN < LN* < i-LN, we can safely believe that Reviewer Msrh would raise the score to Marginally above acceptance.  Reviewer EZi7 is likely to keep the score unchanged, yet the technical concerns should have been resolved by the response. Reviewer SRbU and Reviewer U1uB should keep the score unchanged.

---

### Decision · Program_Chairs · 2026-01-26

Accept (Poster)